# Arbitrage Bounds on Currency Basket Options

**Yi Hong** 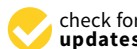

Department of Financial Mathematics, School of Science, Xi'an Jiaotong Liverpool University,
Suzhou 215123, China; yi.hong@xjtlu.edu.cn; Tel.: +8651288161729

**Abstract:** This article exploits arbitrage valuation bounds on currency basket options. Instead of using a sophisticated model to price these options, we consider a set of pricing models that are consistent with the prices of available hedging assets. In the absence of arbitrage, we identify valuation bounds on currency basket options without model specifications. Our results extend the work in the literature by seeking tight arbitrage valuation bounds on these options. Specifically, the valuation bounds are enforced by static portfolios that consist of both cross-currency options and individual options denominated in the numeraire currency.

**Keywords:** currency basket options; static hedging; arbitrage bounds

## 1. Introduction

For many corporations and financial institutions, basket options are an important tool in managing currency exposures. In this article, we derive new results relating the prices of currency basket options to the prices of standard currency option contracts. The need for basket options arises naturally in practice. For example, a company that purchases products from a variety of countries may be exposed to changes in the value of a basket of currencies against its home currency. In seeking to manage its foreign currency exposures, the company could use individual options on each foreign currency separately. But this way would be inefficient when the value changes in one currency are offset by the others to which the company is also exposed. Basket options whose payoffs are based on multiple currency pairs are traded as alternative instruments to manage currency exposures more effectively.

A number of pricing models have been proposed to price and hedge basket options after careful calibration to market prices of options on individual underlying assets. (Although the analytical formula for basket options is unattainable, there exist a number of numerical techniques for pricing and hedging them, for instance, Ashraff, Tarczon and Wu (1995) [1] for a variance-minimizing hedge; Ju (2002) [2] for the method of characterization functions; Brigo, Mercurio, Rapisarda and Scotti (2004) [3] for the moment-matching approach; Pellizzari (2005) [4] for Monte–Carlo simulation; Caldana et al. (2016) [5] for the valuation bounds on basket options for a general class of continuous-time financial models; Bae (2019) [6] for pricing compound basket options.) On the one hand, a wide spectrum of parametric models are available to practitioners. Bates (1996) [7] presents a stochastic volatility jump-diffusion model to explain the skewness and excess kurtosis implicit in currency option prices. Bollen, Gray and Whaley (2000) [8] suggest a regime-switching model and document that the market prices of currency options do incorporate some regime-switching information. Daal and Madan (2005) [9] propose a pure jump model, termed the variance-gamma (VG) model, to capture large movements in exchange rates. Recently, Carr and Wu (2007) [10] and Bakshi, Carr and Wu (2008) [11] develop stochastic skew models to generate both stochastic volatility and stochastic skewness which are documented in currency options. On the other hand, some researchers are interested in copula theory. A copula function is used to construct multivariate density distributions in order to be consistent with the market prices of traded assets. Both Cherubini and Luciano (2002) [12]

and Rosenberg (2003) [13] propose approaches to price basket options with two underlying assets through copula functions.

However, these models are easily mis-specified, because of little information about which is the correct model. A pricing model delivers the precise and fair price for a basket option, only if this model is the true representation of reality. As a result, the method of model building in turn introduces an uncertainty in the choice of model. In this article, we tackle the problem of pricing currency basket options from a different perspective. Rather than using a single parametric model, we consider a set of pricing models that are consistent with the observed prices of traded assets. The aim is to derive model-independent price bounds on currency basket options.

More specifically, a set of currency options is identified as hedging instruments. These assets provide a wide range of hedging strategies for investors. The underlying argument is that vanilla options determine the marginal risk-adjusted probability density of exchange rates, but they do not determine either the complete terminal density or the dynamics of exchange rates. Fitting a model to the prices of vanilla options and using the model for designing a dynamic hedge are subject to errors. Also, it is difficult to perfectly hedge a basket option using option portfolios. These concerns make super-replicating strategies useful. These trading strategies have the appealing features of model independence and simplicity, and they require only static positions in hedging instruments at inception. (The method of super-replication in incomplete financial markets can be linked back to the early studies of Kramkov (1994) [14] and EI Karoui and Quenez (1995) [15].)

Lamberton and Lapeyre (1992) [16] first suggest that basket options could be hedged using portfolios of underlying assets. Bertsimas and Popescu (2002) [17] investigate the super-replication of financial derivatives, including basket options and other exotics. Given the knowledge about the moments of return distributions or the prices of relevant hedging assets, they propose a convex optimization method to derive valuation bounds on complex financial derivatives. This method is closely related to the techniques developed by Gotoh and Konno (2002) [18], who propose an efficient algorithm to deal with semidefinite programmes in order to attain valuation bounds on basket options.

D'Aspremont and EI-Ghaoui (2003) [19] and Pena, Vera and Zuluaga (2006) [20] apply the linear programming (LP) approach to price basket options and suggest that valuation bounds on these options can be derived from the prices of other relevant basket options. Laurence and Wang (2005) [21] investigate the relation between pricing and hedging basket options. In these three papers, valuation bounds on basket options with two underlying assets may be expressed analytically. In particular, Laurence and Wang (2005) [21] assume that there is only one strike for each individual option.

In Hobson, Laurence and Wang (2005a, 2005b) [22,23], the arbitrage bounds on basket options in a general setup are derived using portfolios of options on individual underlying assets with the number of $n \geq 2$. These studies extend the results of Laurence and Wang (2005) [21] when traded options are available at a continuum of strike prices. In the first paper, they use a Lagrange optimization approach to characterize the optimal strikes. A super-replicating strategy that enforces an upper bound is simply a linear combination of European call options. To support upper price bounds, underlying asset processes must be comonotonic. In the second paper, they construct the so-called "STP" portfolios to sub-replicate basket options. Countermonotonic underlying processes yield lower price bounds. Chen, Deelstra, Dhaene and Vanmaele (2008) [24] construct static super-replicating strategies for a class of exotic options written on a weighted sum of asset prices, including Asian options and basket options, among others. Based on the theory of integral stochastic orders, they provide a characterization for the optimal strikes, which is different from the methodology proposed by Hobson, Laurence and Wang (2005a) [22].

Slightly different from the setup in Hobson, Laurence and Wang (2005a, 2005b) [22,23], this study explores the arbitrage valuation bounds of currency basket options in the presence of both cross-currency options and individual options quoted in a dominant currency. In terms of pricing basket options on three currencies (i.e., the Euro, British pound and U.S. dollar), we make use of the facts that (i) there typically exist deep and liquid markets in these three currency pairs;

(ii) the prices of cross-currency options are actually attainable (by contrast such options are usually unavailable in equity markets); and (iii) the prices of these options carry useful information about the joint distribution of underlying currencies. We find that the valuation bounds on currency basket options could be further tightened when the cross-currency options are incorporated. These valuation bounds are also enforced by static portfolios of both cross-currency options and options denominated in the numeraire currency.

This article is organized as follows. Section 2 introduces the setup. The properties of both the marginal and joint densities of underlying currency pairs are discussed. Section 3 presents main results. Section 4 provides a numerical analysis regarding both dominating (dominated) strategies and joint distributions. The final section concludes.

## 2. Preliminaries

Consider a single-period setting in a frictionless currency market (i.e., no short sale restrictions, transaction costs and other frictions). Within this setup, all investments are made at time zero, and all payments are received at time $T$. There are three main currencies, the Euro (EUR, €), British pound (GBP, £) and U.S. dollar (USD, $). The interest rates in all currencies are zero. Let the dollar ($) be the numeraire currency. The positive variables $X$ and $Y$ represent the (unknown) dollar-denominated prices of the Euro and British pound at maturity,

$$X, Y \in \mathbb{R}^+. \tag{1}$$

Their time-0 prices are $x_0$ ($> 0$) and $y_0$ ($> 0$). We hereafter also use them to indicate the corresponding foreign currencies unless otherwise specified. Let the variable $Z = Y/X$ represent the Euro-denominated price of the Pound at maturity.

There are European-style call options written on these three currency pairs at all strikes. It is assumed that the prices of these options are twice differentiable, convex and decreasing with respect to strike. Specifically, there exist two complete sets of dollar-denominated options on the Euro ($X$) and Pound ($Y$). There also exists a complete set of cross-currency options, the $X$-denominated options on $Y$. All these options mature at time $T$. Put options with the same maturity on individual currency pairs are known through the call–put parity.

Within this setup, we make the following assumption:

[A1] There is no arbitrage among all hedging assets.

Since a continuum of call option prices is available and these prices are twice differentiable, Breeden and Litzenberger (1978) [25] establish that the pricing density of Arrow–Debreu claims can be inferred from option prices. Therefore, the available option prices imply that for each of three exchange rates there exists a state price density. Since option prices are twice differentiable, each price density is continuous with respect to strike.

Let $\pi^i(k)$ ($i = X, Y, Z$) be the price density of an Arrow–Debreu claim that pays 1 unit domestic currency if an exchange rate reaches the level $k$ and zero otherwise. Similarly, let the integrable function $p(x, y)$ be the price density (or pricing function) of a claim that pays $1 at maturity if $X = x$ and $Y = y$. Since the interest rates in all currencies are zero, the pricing function $p$ has two properties: (1) $\int_{\mathbb{R}_2^+} p(x, y) dx dy = 1$ and (2) $p(x, y) \geq 0$. Let $\mathcal{P}$ be the set of all pricing functions. Assumption [A1] implies that the set $\mathcal{P}$ is not empty.

**Lemma 1.** *Given the dollar-denominated options on X and Y and the X-denominated options on Y, the absence of arbitrage implies the following equalities*

$$1)\ \pi^X(x) = \int_{\mathbb{R}^+} p(x, y) dy;\ 2)\ \pi^Y(y) = \int_{\mathbb{R}^+} p(x, y) dx;\ 3)\ \pi^Z(z) = \frac{1}{x_0}\left(\int_{\mathbb{R}^+} x p(x, xz) dx\right), \tag{2}$$

The following lemma shows that the similar properties in Lemma 1 are maintained if the currency base is changed.

**Lemma 2.** *Given a pricing function $p \in \mathcal{P}$ that satisfies the conditions in (2), then a new pricing function $p_X$ in the currency base X can be derived from the function p in the following way:*

$$p_X(x,y) = \frac{1}{x_0 x} p(\frac{1}{x}, \frac{y}{x}), \text{ for } (x,y) \in \mathbb{R}_2^+,$$

*such that*

$$1) \int_{\mathbb{R}^+} p_X(x,y) dy = \frac{\pi^X(\frac{1}{x})}{x_0 x}; 2) \int_{\mathbb{R}^+} p_X(x,y) dx = \pi^Z(y); 3) \int_{\mathbb{R}^+} x p_X(x,xz) dx = \frac{\pi^Y(z)}{x_0}. \quad (3)$$

Throughout this study, we are interested in the valuation bounds on a basket call option that delivers the dollar-denominated payoff (The function $(x)^+$ takes the non-negative part of $x$.)

$$(\alpha X + \beta Y - \gamma)^+, \text{ for } (\alpha, \beta, \gamma) \in \mathbb{R}^3. \quad (4)$$

Note that this representation also contains the payoffs of spread options. As far as the valuation bounds on this call is attained, the "put-call" parity relationship for European-style basket options, as pointed out by Hobson, Laurence and Wang (2005a, 2005b) [22,23] and Su (2005) [26], immediately implies that the model-independent valuation bounds on a basket put can be obtained:

$$(\gamma - (\alpha X + \beta Y))^+ = (\alpha X + \beta Y - \gamma)^+ - (\alpha X + \beta Y - \gamma). \quad (5)$$

As for the payoff in (4), state price densities implied from option prices impose restrictions on a set of pricing functions. The possible value range of this payoff is determined by all pricing functions in $\mathcal{P}$ that satisfy the conditions in (2). We now establish valuation bounds on call options in (4).

## 3. Valuation Bounds and Hedging Portfolios

We seek arbitrage valuation bounds. When dollar-denominated options are traded as hedging instruments, the results in Hobson, Laurence and Wang (2005a, 2005b) [22,23] show that valuation bounds on the payoff in (4) ($\alpha, \beta, \gamma > 0$) can be attained. Our main result is that valuation bounds are (much) tightened when cross-currency options are traded as hedging instruments. Valuation bounds are enforced by static portfolios of dollar-denominated options and cross-currency options. Also, the pricing function that maximizes a basket option's price is characterized.

We at present look only at upper price bounds, and lower bounds will be discussed later. This section first formulates the valuation problem of a basket option in (4) as an infinite-dimensional LP. This problem is to find the maximum price bound on this basket option within a set of pricing functions. These functions are subject to restrictions imposed by option prices. The dual problem is to search for dominating strategies which enforce upper price bounds. These strategies ensure that an agent who writes this option can put a floor on potential losses.

### 3.1. Problem Formulation

Consider the basket option that delivers the payoff in (4) at maturity. Its dollar-denominated price is formally expressed as follows:

$$\mathbb{E}_p[(\alpha X + \beta Y - \gamma)^+] = \int_{\mathbb{R}_2^+} [\alpha x + \beta y - \gamma]^+ p(x,y) dx dy, \text{ for } p \in \mathcal{P}. \quad (6)$$

Hence, the price bound on this option is attained by seeking all pricing functions over the entire set $\mathcal{P}$.

To seek the least upper price bound, we express the valuation problem as an LP:

$$\max_{p \in \mathcal{P}} \int_{\mathbb{R}_2^+} [\alpha x + \beta y - \gamma]^+ p(x,y) dx dy \tag{7}$$

s.t.

$$1) \int_{\mathbb{R}^+} p(x,y) dy = \pi^X(x); 2) \int_{\mathbb{R}^+} p(x,y) dx = \pi^Y(y); 3) \int_{\mathbb{R}^+} \frac{x}{x_0} p(x, xz) dx = \pi^Z(z).$$

The initial market prices of options are incorporated into the three constraints. Furthermore, the first two constraints ensure the following

$$\int_{\mathbb{R}_2^+} p(x,y) dx dy = 1.$$

The feasible set of this program is not empty due to assumption [A1]. The value of this program is bounded below by zero and from above:

$$\begin{aligned} \mathbb{E}_p[(\alpha X + \beta Y - \gamma)^+] &= \mathbb{E}_p[(\alpha(X - k_1) + \beta(Y - k_2) + (\alpha k_1 + \beta k_2 - \gamma))^+] \\ &\le \mathbb{E}_p[(\alpha(X - k_1))^+] + \mathbb{E}_p[(\beta(Y - k_2))^+] + \mathbb{E}_p[(\alpha k_1 + \beta k_2 - \gamma)^+] < \infty, \end{aligned} \tag{8}$$

for any strikes $k_1, k_2 \in \mathbb{R}^+$. So the program in (7) must have a solution.

The dual of the problem in (7) is to find the cheapest dominating strategy. Let $\phi = (g(x), h(y), f(z))$ $(x, y, z \in \mathbb{R}^+)$ be a trading strategy such that the functions $g, h$ and $f$ represent the respective components of dollar-denominated and cross-currency options. As a result, the hedging problem may be described as follows:

$$\min_{g,h,f \in \mathbb{R}} \int_{\mathbb{R}^+} g(x) \pi^X(x) dx + \int_{\mathbb{R}^+} h(y) \pi^Y(y) dy + x_0 \int_{\mathbb{R}^+} f(z) \pi^Z(z) dz, \tag{9}$$

s.t.

$$1') g(x) + h(y) + x f(z) \mathbf{1}_{z=y/x} \ge (\alpha x + \beta y - \gamma)^+ \text{ for all } x, y, z \in \mathbb{R}^+,$$

where $\mathbf{1}_{(\cdot)}$ is an indicator function. Both programs have the same Lagrangian form. Their equivalence is established through the following result:

**Proposition 1** (Strong Duality)**.** *Given assumption [A1], the values of the primal in (7) and the dual in (9) coincide.*

This strong duality follows from Isii (1963) [27], Gotoh and Konno (2002) [18] and Laurence and Wang (2005) [21]. The necessary condition required in Isii's theorem is satisfied in (8):

$$\mathbb{E}_p[(\alpha X + \beta Y - \gamma)^+] < \infty, \text{ for all } p \in \mathcal{P}.$$

Hence, there exists a strategy which involves trading dollar-denominated options and cross-currency options. All the strategies that solve the program in (9) construct a non-empty set $\mathcal{A}$.

Lemma 2 shows that if the currency base is changed, a new pricing function can be constructed from a pricing function that solves the program in (7). This new pricing function also maximizes the basket option's price in the new currency base, and the associated trading strategy is a dominating one.

**Proposition 2.** *Given the dollar as the currency base, suppose that there exists a pair $(p, \phi)$ $(p \in \mathcal{P}, \phi \in \mathcal{A})$ that supports the market prices of all traded options. If $(p, \phi)$ is optimal for the programs in (7) and (9) respectively, so is $(p_X, \phi)$ based on the currency X.*

This proposition establishes the correspondence between pricing and hedging basket options in different currency bases. Dominating strategies do not depend on the choice of base currency. In the next section, we first establish upper bounds on basket options in the case where only

dollar-denominated options are tradable. Valuation bounds on currency basket options with two underlying assets may be sought by solving the primal problem where the third restriction in (7) is dropped. Trading strategies that enforce these bounds can be sought by setting $f(z) \equiv 0$ in (9). We then assume that cross-currency options are also traded in markets, and show how these options can tighten valuation bounds.

## 3.2. Upper Valuation Bounds

When only the dollar-denominated options on $X$ and $Y$ are available, the upper price bounds on a basket option with positive parameters ($\alpha, \beta, \gamma > 0$) can be attained by applying the result in Hobson, Laurence and Wang (2005a) [22]. These bounds are enforced by static portfolios of dollar-denominated options.

**Proposition 3.** *Given a triplet* $(\alpha, \beta, \gamma) \in \mathbb{R}_3^+$, *the upper price bounds on the option* $\boldsymbol{b}$ *in (4) are achieved by the dollar-denominated options on X and Y as follows:*

$$\min_{K_a, K_b \geq 0: \alpha K_a + \beta K_b = \gamma} \alpha \int_{\mathbb{R}^+} (x - K_a)^+ \pi^X(x) dx + \beta \int_{\mathbb{R}^+} (y - K_b)^+ \pi^Y(y) dy. \tag{10}$$

*The associated pricing function* $p \in \mathcal{P}$ *is characterized as follows:*

$$p^*(x, y) = \begin{cases} \geq 0, & \text{if } (x, y) \in (0, K_a^*] \times (0, K_b^*]; \\ \geq 0, & \text{if } (x, y) \in (K_a^*, \infty) \times (K_b^*, \infty); \\ 0, & \text{otherwise,} \end{cases} \tag{11}$$

*where the strikes* $K_a^*$ *and* $K_b^*$ *solve the problem (10).*

From Hobson, Laurence and Wang (2005a) [22], the cheapest dominating strategy is sought via a Lagrangian approach. As shown in (10), dominating strategies are to buy call options on the Euro with strike $K_a$ and call options on the Pound with strike $K_b$. For a basket option in (4) ($\alpha, \beta, \gamma > 0$), the strikes are chosen so that in the region where both calls are in the money, and two sets of options replicate it exactly. There is no possibility of one option being in the money and the other being out of the money. Since there is no assumption on the behavior of currency prices, these dominating strategies are robust to both model and correlation misspecification.

On the other hand, the dollar-denominated options indeed provide information about the joint distribution of the variables $X$ and $Y$ at maturity. Conditional on the marginal price densities, the joint density in (11) that maximizes the value of the basket option $\boldsymbol{b}$ suggests that the variables $X$ and $Y$ are strongly correlated. The left panel in Figure 1 illustrates this pricing function. Since the basket option is an option on a basket of the Euro (X) and Pound (Y), maximizing the correlation between $X$ and $Y$ ensures the maximum volatility for the basket and hence the maximum value for an option on the basket.

If the $X$-denominated options on $Y$ are traded, information embedded in these options tends to restrict the range of correlation between $X$ and $Y$. The following statement establishes that valuation bounds on basket options are enforced by static portfolios that consist of both dollar-denominated and cross-currency options.

**Proposition 4.** *Given a triplet* $(\alpha, \beta, \gamma) \in \mathbb{R}_3^+$, *the upper price bounds on the option* $\boldsymbol{b}$ *in (4) are achieved by the dollar-denominated options on X and Y and the X-denominated options on Y as follows:*

$$\min_{\lambda,\delta,z_1,z_2,K_i,i=1,2,3,4} \int_{\mathbb{R}^+} [(\alpha - \lambda)(K_1 - x)^+ + \lambda(x - K_2)^+$$
$$+ (\alpha - \lambda)(x - K_3)^+ + \lambda(x - K_4)^+]\pi^X(x)dx$$
$$+ \int_{\mathbb{R}^+} [(\beta - \delta)(z_1 K_2 - y)^+ + \delta(y - z_2 K_1)^+ \qquad (12)$$
$$+ (\beta - \delta)(y - z_1 K_4)^+ + \delta(y - z_2 K_3)^+]\pi^Y(y)dy$$
$$+ \int_{\mathbb{R}^+} [-(\beta - \delta)(z_1 - z)^+ + (-\delta)(z - z_2)^+]\pi^Z(z)dz$$

*where*

1)$(\beta - \delta)z_1 = \lambda; \delta z_2 = \alpha - \lambda$, *for* $0 \leq z_1 \leq \frac{\alpha}{\beta} \leq z_2$ *and* $\lambda\delta \neq 0$;

2)$\lambda(K_3 - K_2) + \delta(z_1 K_4 - z_2 K_1) = \alpha K_3 + \beta z_1 K_4 - \gamma$;

3)$0 \leq \lambda < \alpha; 0 \leq \delta < \beta; 0 \leq K_1 \leq K_2 \leq K_3 \leq K_4$.

*The associated probability density function* $p \in \mathcal{P}$ *is characterized as follows:*

$$p^*(x,y) = \begin{cases} \geq 0, & if\ (x,y) \in (0, K_1^*] \times [z_2^* K_1^*, z_1^* K_4^*] \cup [K_1^*, K_2^*] \times [z_1^* K_2^*, z_2^* K_1^*]; \\ \geq 0, & if\ (x,y) \in [K_2^*, K_3^*] \times (0, z_1^* K_2^*] \cup [K_2^*, K_3^*] \times [z_2^* K_3^*, \infty); \\ \geq 0, & if\ (x,y) \in [K_3^*, K_4^*] \times [z_1^* K_4^*, z_2^* K_3^*] \cup [K_4^*, \infty) \times [z_2^* K_1^*, z_1^* K_4^*]; \\ = 0, & otherwise, \end{cases} \qquad (13)$$

*where the strikes* $K_i^*$ *(*$i = 1,2,3,4$*),* $z_1^*, z_2^*$ *and real numbers* $\lambda^*, \delta^*$ *solve the problem* (12).

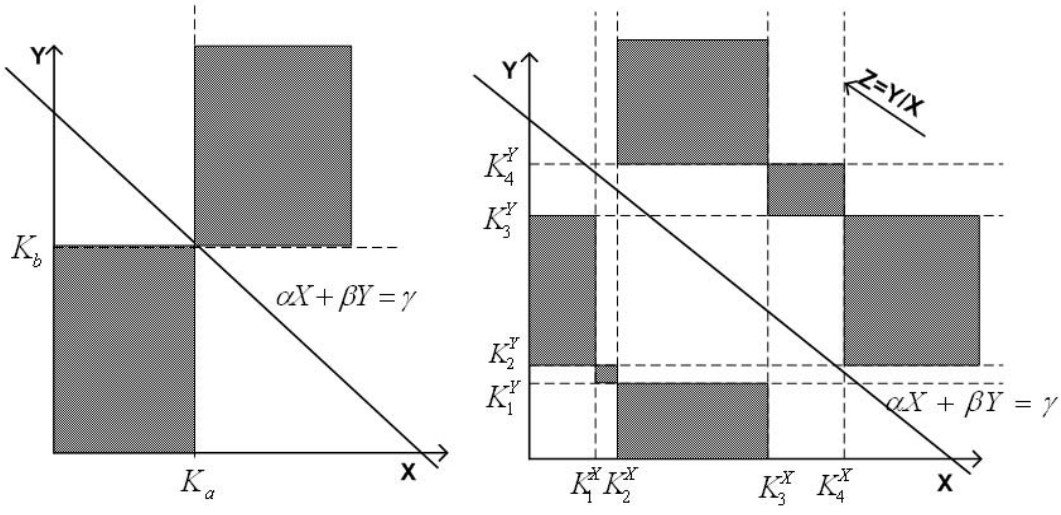

**Figure 1. Joint Density Distributions for Upper Bounds**. The left panel shows a price density (pricing function) $p(x,y)$ that maximizes the price of the option **b** if only dollar-denominated options are traded. The right panel illustrates a price density that maximizes this option's price when cross-currency options are traded as hedging instruments. The shaded parts indicate the regions where the density $p$ is non-negative. In the right panel, it has $(K_1^Y, K_2^Y, K_3^Y, K_4^Y) = (z_1 K_2^X, z_2 K_1^X, z_1 K_4^X, z_2 K_3^X)$ for $0 < z_1 < z_2$.

To dominate the option **b**, hedging strategies involve eight variables. Four variables $K_i$ ($i = 1,2,3,4$) are used to determine the specific price levels on the Euro ($X$). These variables together with two variables $z_1$ and $z_2$ determine the corresponding price levels on the Pound ($Y$). These price levels are strikes for buying or selling dollar-denominated options or cross-currency

options. The quantity of each hedging instrument is specified by the variables $\lambda$ and $\delta$. There exists a feasible solution which is identified in (12):

$$
\begin{aligned}
(K_1, K_2, K_3, K_4, z_1, z_2) &= (0, K_a, K_a, \infty, 0, \infty); \\
(z_1 K_2, z_2 K_1, z_1 K_4, z_2 K_3, \lambda, \delta) &= (0, K_b, K_b, \infty, 0, 0),
\end{aligned}
\tag{14}
$$

where $\alpha K_a + \beta K_b = \gamma$. Note that the equalities in the first constraint are valid only for $\lambda \in (0, \alpha)$ and $\delta \in (0, \beta)$, and hence the feasible solution above is consistent with the constraints in (12). This program therefore has a solution.

In the presence of only dollar-denominated options, a bivariate process $(X, Y)$ that maximizes a basket option's price implies the strong dependence between two currency pairs. By incorporating information about cross-currency options, valuation bounds on basket options can be tightened. As a result, the pricing function in (13) that also maximizes the basket option' price indicates that the strong dependence two currency pairs might be unnecessary due to restrictions on correlation between them imposed by cross-currency options. The right panel in Figure 1 illustrates this pricing function.

Through Propositions 3 and 4, the upper bounds on the option **b** are derived for positive parameters $(\alpha, \beta, \gamma > 0)$. However, it is unnecessary to require that a currency basket is constructed by only buying two currencies and selling another. This restriction is relaxed through the following statement.

**Proposition 5.** *Given a triplet $(\alpha, \beta, \gamma) \in \mathbb{R}_3^-$, the upper price bounds on the option **b** in (4) are achieved by the dollar-denominated options on X and Y as follows:*

$$
\min_{K_a, K_b \geq 0 : \alpha K_a + \beta K_b = \gamma} (-\alpha) \int_{\mathbb{R}^+} (K_a - x)^+ \pi^X(x) dx + (-\beta) \int_{\mathbb{R}^+} (K_b - y)^+ \pi^Y(y) dy.
\tag{15}
$$

*If the X-denominated options on Y are traded, the bounds are further enforced as follows:*

$$
\begin{aligned}
\min_{\lambda, \delta, z_1, z_2, K_i, i=1,2,3,4} \int_{\mathbb{R}^+} & [\lambda(K_1 - x)^+ + (-\alpha - \lambda)(K_2 - x)^+ \\
& + \lambda(K_3 - x)^+ + (-\alpha - \lambda)(x - K_4)^+] \pi^X(x) dx \\
& + \int_{\mathbb{R}^+} [\delta(z_1 K_2 - y)^+ + (-\beta - \delta)(z_2 K_1 - y)^+ \\
& + \delta(z_1 K_4 - y)^+ + (-\beta - \delta)(z_2 K_3 - y)^+] \pi^Y(y) dy \\
& + \int_{\mathbb{R}^+} [(-\delta)(z_1 - z)^+ + (\beta + \delta)(z - z_2)^+] \pi^Z(z) dz
\end{aligned}
\tag{16}
$$

*where*

$$
\begin{aligned}
&1) \delta z_1 = (-\alpha) - \lambda; (-\beta - \delta) z_2 = \lambda, \text{ if } 0 \leq z_1 \leq \tfrac{\alpha}{\beta} \leq z_2 \text{ and } \lambda \delta \neq 0; \\
&2) \lambda(K_3 - K_2) + \delta(z_1 K_4 - z_2 K_1) = \alpha K_2 + \beta z_2 K_1 - \gamma; \\
&3) 0 \leq \lambda < (-\alpha); 0 \leq \delta < (-\beta); 0 \leq K_1 \leq K_2 \leq K_3 \leq K_4.
\end{aligned}
$$

The proof of this proposition and the characterization of price functions are accomplished similarly, according to Propositions 3 and 4. In fact, upper valuation bounds on the option **b** in (4) are derived from these results, which will be discussed later.

### 3.3. Lower Valuation Bounds

We have derived upper valuation bounds on currency basket options. Similarly, lower bounds on these options can be attained by solving LPs. From Hobson, Laurence and Wang (2005b) [23], lower price bounds on basket options with two underlying assets are enforced by the so-called "STP" portfolios that involve calls and puts on individual underlying assets. The associated bivariate

processes $(X, Y)$ should be counter-monotonic. Nevertheless, their lower valuation bounds are dependent on the number of disjoint intervals over $\mathbb{R}^+$. We establish a lemma to simplify their result.

**Lemma 3.** *Suppose that $\mathbb{R}^+$ is partitioned into $(2n + 1)$ $(n \geq 1)$ disjoint intervals. Given a triplet $(\alpha, \beta, \gamma) \in \mathbb{R}_3^+$, the lower bounds attained by Hobson, Laurence and Wang (2005b) [23] are the non-increasing functions of partition number $(n)$.*

This lemma shows that the greatest lower bound is determined by setting $n = 1$. As a result, sub-replicating strategies are simplified.

**Proposition 6.** *(1) Given a triplet $(\alpha, \beta, \gamma) \in \mathbb{R}_3^+$, the lower price bounds on the option **b** in (4) are given by the dollar-denominated options on X and Y as follows:*

$$\max_{0 < K_a^1 \leq K_a^2, 0 < K_b^1 \leq K_b^2} \int_{\mathbb{R}^+} ((-\alpha)(K_a^1 - x)^+ + \alpha(x - K_a^2)^+) \pi^X(x) dx$$
$$+ \int_{\mathbb{R}^+} ((-\beta)(K_b^1 - y)^+ + \beta(y - K_b^2)^+) \pi^Y(y) dy, \tag{17}$$

*where $\alpha K_a^1 + \beta K_b^2 = \alpha K_a^2 + \beta K_b^1 = \gamma$.*
*(2) Given a triplet $(\alpha, \beta, \gamma) \in \mathbb{R}_3^-$, the lower price bounds on this option are attained as follows:*

$$\max_{0 < K_a^1 \leq K_X \leq K_a^2, 0 < K_b^1 \leq K_Y \leq K_b^2} \int_{\mathbb{R}^+} (\alpha(K_a^1 - x)^+ + (-\alpha)(K_X - x)^+$$
$$+ \alpha(x - K_X)^+ + (-\alpha)(x - K_a^2)^+) \pi^X(x) dx$$
$$+ \int_{\mathbb{R}^+} (\beta(K_b^1 - y)^+ + (-\beta)(K_Y - y)^+$$
$$+ \beta(y - K_Y)^+ + (-\beta)(y - K_b^2)^+) \pi^Y(y) dy, \tag{18}$$

*where $\alpha K_a^1 + \beta K_b^2 = \alpha K_a^2 + \beta K_b^1 = \alpha K_X + \beta K_Y = \gamma$.*
*The associated price density function $p \in \mathcal{P}$ is characterized as follows:*

$$p^*(x, y) = \begin{cases} \geq 0, & \text{if } (x, y) \in (0, \hat{K}_a^1] \times [\hat{K}_b^2, \infty) \text{ for } |\alpha|x + |\beta|y \geq |\gamma|; \\ \geq 0, & \text{if } (x, y) \in [\hat{K}_a^1, \hat{K}_a^2] \times [\hat{K}_b^1, \hat{K}_b^2] \text{ for } |\alpha|x + |\beta|y \leq |\gamma|; \\ \geq 0, & \text{if } (x, y) \in [\hat{K}_a^2, \infty) \times (0, \hat{K}_b^1] \text{ for } |\alpha|x + |\beta|y \geq |\gamma|; \\ 0, & \text{otherwise,} \end{cases} \tag{19}$$

*where the strikes $\hat{K}_a^1$, $\hat{K}_a^2$, $\hat{K}_b^1$ and $\hat{K}_b^2$ solve the problems (17) or (18).*

The first part of Proposition 6 directly comes from Lemma 3. Dominating strategies involve short selling puts and long buying calls on the Euro (Pound) with the strikes $K_a^1$ ($K_b^1$) and $K_a^2$ ($K_b^2$) so that in the regions both calls and puts are in the money and two sets of options replicate the option **b** exactly. Sub-replicating strategies are slightly different when all the parameters in (4) are negative, and they involve trading more calls and puts. Like dominating strategies specified in Proposition 3, dominated strategies identified here are robust to both model and correlation mis-specification. Meanwhile, the dual provides information about the joint density of the variables $X$ and $Y$ at maturity. In order to minimize the option **b**'s price, the price density function $p$ in (19) suggests that the process $(X, Y)$ should be counter-monotonic. The left panel of Figure 2 illustrates this pricing function.

Now we derive tight valuation bounds when cross-currency options are traded. These valuation bounds are also enforced by static portfolios that consist of both dollar-denominated options and cross-currency options. Yet, the process $(X, Y)$ that minimizes the option **b**'s price might not be counter-monotonic.

**Proposition 7.** *(1) Given a triplet $(\alpha, \beta, \gamma) \in \mathbb{R}_3^+$, the lower price bounds on the option* **b** *in (4) are achieved by the dollar-denominated options on X and Y and the X-denominated options on Y as follows:*

$$
\max_{\lambda, \delta, z_1, z_2, K_i, i=1,2,3,4} \int_{\mathbb{R}^+} (\lambda(x - K_3)^+ + (\alpha - \lambda)(x - K_4)^+) \pi^X(x) dx
$$
$$
+ \int_{\mathbb{R}^+} (\delta(y - z_2 K_1)^+ + (\beta - \delta)(y - z_2 K_2)^+) \pi^Y(y) dy \tag{20}
$$
$$
+ \int_{\mathbb{R}^+} ((-\beta)(z_1 - z)^+ + (\beta - \delta)(z - z_2)^+) \pi^Z(z) dz,
$$

*where*

> 1) $\beta z_1 = \lambda - \alpha; (\delta - \beta) z_2 = \alpha, \text{for } 0 \le z_1 \le z_2;$
> 2) $\lambda(K_4 - K_3) + \delta(z_2 K_2 - z_2 K_1) = \alpha K_4 + \beta z_2 K_2 - \gamma;$
> 3) $\alpha K_1 + \beta z_2 K_1 = \gamma; \alpha K_3 + \beta z_1 K_3 = \gamma;$
> 4) $\lambda \ge \alpha; \delta \ge \beta; 0 \le K_1 \le K_2 \le K_3 \le K_4.$

*(2) Given a triplet $(\alpha, \beta, \gamma) \in \mathbb{R}_3^-$, the lower price bounds on this option are produced as follows:*

$$
\max_{\lambda, \delta, z_1, z_2, K_i, i=1,2,3,4} \int_{\mathbb{R}^+} ((\lambda - \alpha)(K_3 - x)^+ + (-\lambda)(K_4 - x)^+) \pi^X(x) dx
$$
$$
+ \int_{\mathbb{R}^+} ((\delta - \beta)(z_2 K_1 - y)^+ + (-\delta)(z_2 K_2 - y)^+) \pi^Y(y) dy \tag{21}
$$
$$
+ \int_{\mathbb{R}^+} (\beta(z_1 - z)^+ + (-\delta)(z - z_2)^+) \pi^Z(z) dz,
$$

*where*

> 1) $\beta z_1 = -\lambda; \delta z_2 = -\alpha, \text{for } 0 \le z_1 \le z_2;$
> 2) $\lambda(K_3 - K_4) + \delta(z_2 K_1 - z_2 K_2) = \alpha K_3 + \beta z_2 K_1 - \gamma;$
> 3) $\alpha K_1 + \beta z_2 K_1 = \gamma; \alpha K_3 + \beta z_1 K_3 = \gamma;$
> 4) $\lambda \ge 0; \delta \ge 0; 0 \le K_1 \le K_2 \le K_3 \le K_4.$

*The associated price function $p \in \mathcal{P}$ is characterized as follows:*

$$
p^*(x,y) = \begin{cases}
\ge 0, & \text{if } (x,y) \in (0, \hat{K}_3] \times (0, \hat{z}_2 \hat{K}_1] \text{ for } \frac{y}{x} \in [\hat{z}_1, \hat{z}_2] \text{ and } |\alpha|x + |\beta|y \le |\gamma|; \\
\ge 0, & \text{if } (x,y) \in (0, \hat{K}_2] \times [\hat{z}_2 \hat{K}_1, \hat{z}_2 \hat{K}_2] \text{ for } \frac{y}{x} \ge \hat{z}_2 \text{ and } |\alpha|x + |\beta|y \ge |\gamma|; \\
\ge 0, & \text{if } (x,y) \in [\hat{K}_3, \hat{K}_4] \times (0, \hat{z}_1 \hat{K}_4] \text{ for } \frac{y}{x} \le \hat{z}_1 \text{ and } |\alpha|x + |\beta|y \ge |\gamma|; \\
\ge 0, & \text{if } (x,y) \in [\hat{K}_4, \infty) \times [\hat{z}_2 \hat{K}_2, \infty) \text{ for } \frac{y}{x} \in [\hat{z}_1, \hat{z}_2]; \\
0, & \text{otherwise,}
\end{cases}
$$

*where the strikes $\hat{K}_1, \hat{K}_2, \hat{K}_3, \hat{K}_4, \hat{z}_1$ and $\hat{z}_2$ solve the problems (20) or (21).*

The existence of the strikes that satisfy the conditions in Proposition 7 imply that there at least exits one dominated trading strategy that consists of dollar-denominated options and cross-currency options. This strategy is independent of model specification. Note that the strategies identified in Proposition 6 can be viewed as particular cases of dominated strategies specified in Proposition 7 for $z_1 \to 0$ and $z_2 \to \infty$. Meanwhile, the joint density of the process $(X, Y)$ that minimizes the options **b**'s price is shown in the right panel of Figure 2.

Since the strategies identified in Proposition 6 can be viewed as the portfolios without the cross-currency options in an augmented hedging instrument set, the bounds derived from the program in (20) or (21) should not be cheaper than the bounds attained from program in (17) or (18). Furthermore,

we now establish the following statement that both the upper and lower valuation bounds on the option **b** with the general payoff are attainable.

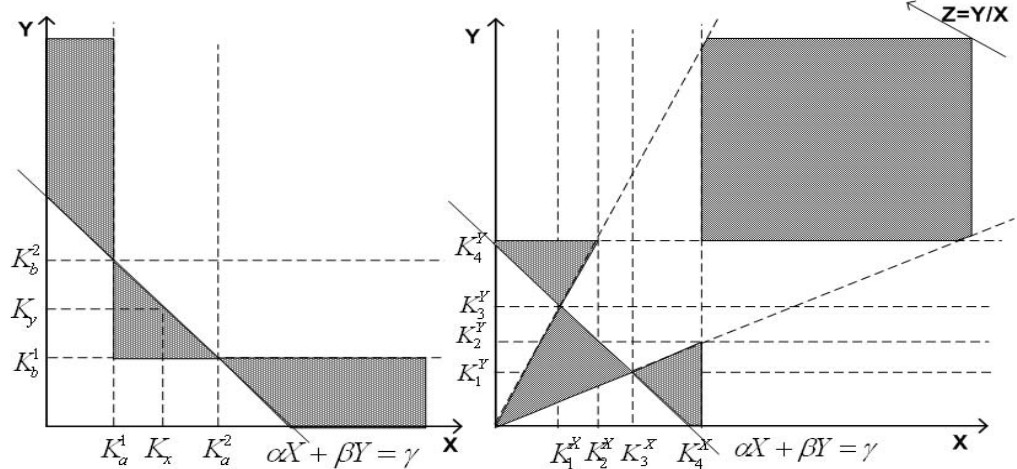

**Figure 2. Joint Density Distributions for Lower Bounds**. The left panel shows a price density (pricing function) $p(x,y)$ that minimizes the price of the option **b** if only dollar-denominated options are traded. The right panel illustrates a price density that minimizes this option's price when cross-currency options are traded as hedging instruments. The shaded parts indicate the regions where the density $p$ is non-negative. In the right panel, it has $(K_1^Y, K_2^Y, K_3^Y, K_4^Y) = (z_1 K_3^X, z_1 K_4^X, z_2 K_1^X, z_2 K_2^X)$.

**Theorem 1.** *Given a triplet* $(\alpha, \beta, \gamma) \in \mathbb{R}_3$, *the upper valuation bounds on the option* **b** *in* (4) *can be derived from Propositions* 3, 4 *and* 5, *while the lower valuation bounds are attained through Propositions* 6 *and* 7.

**Proof.** The signs of $\alpha, \beta, \gamma$ have the following possible combinations:

| ♯ | $\alpha$ | $\beta$ | $(-\gamma)$ | ♯ | $\alpha$ | $\beta$ | $(-\gamma)$ |
|---|---|---|---|---|---|---|---|
| 1 | + | + | + | 5 | + | - | - |
| 2 | + | + | - | 6 | - | - | + |
| 3 | + | - | + | 7 | - | + | - |
| 4 | - | + | + | 8 | - | - | - |

For ♯1 and ♯8, they are degenerate in the sense that the price of the option **b** is always (♯1) or never (♯8) in the money. We have sought valuation bounds in ♯2 (Propositions 3 and 4) and ♯6 (Proposition 5). Valuation bounds in ♯3 and ♯4 can be attained from Proposition 3 in the appropriate currency bases. Similarly, Proposition 5 can be applied to ♯5 and ♯7 by changing currency bases. Similarly, the lower valuation bounds then can be attained from Propositions 6 and 7.  □

## 4. Numerical Analysis

Given the prices of options on three currency pairs, we now numerically investigate arbitrage bounds on basket options, and put the problem into a discrete setup. Within this setup, the prices of call options on all currency pairs are generated and accordingly three price densities are constructed. From these price densities, a numerical procedure for seeking both upper bounds and dominating strategies is proposed. Finally, we qualify the tightness of valuation bounds, and investigate their sensitivity to relevant parameters.

### 4.1. Model Implementation

Suppose that the variables $X$ and $Y$ take values in the sets

$$X \in \{x_0 u^n | n \in \mathbb{N}\},$$
$$Y \in \{y_0 u^n | n \in \mathbb{N}\}, \tag{22}$$

for three positive real numbers $x_0, y_0$, $u > 1$, and $\mathbb{N} = \{-N, \cdots, +N\}$ (an integer number $N$). The variable $Z$ is determined by $Z = Y/X$.

Now consider a basket option **b** that pays $[\alpha X + \beta Y - \gamma]^+$ dollars at maturity. In this finite-state model, the problems presented in Section 3.1 are naturally expressed as finite-dimensional LPs. Let a $(2N+1) \times (2N+1)$ matrix $P = \{p_{m,n}\}$ represent a pricing function. Let two $(2N+1)$ vectors $\{\pi_m^X\}$ and $\{\pi_n^Y\}$ $(m, n \in [-N, N])$ represent the price densities of Arrow–Debreu claims implied from the dollar-denominated options on $X$ and $Y$. To ensure that all three price densities are consistent in scale, a restriction on $P$ is imposed so that $p_{m,n} = 0$ if $|m - n| > N$. For the price density of Arrow–Debreu claims implied from cross-currency options, represented by a vector $\{\pi_j^Z\}$, this restriction equivalently states $\pi_{j=m-n}^Z \geq 0$ for $|m - n| \leq N$ and zero otherwise.

To seek the least upper bound on this basket option, the primal problem in (7) can be reexpressed as a finite LP

(LP1) find the $(2N+1) \times (2N+1)$ matrix $P$ so that
$$\max_P \sum_{m,n} (\alpha x_0 u^m + \beta y_0 u^n - \gamma)^+ p_{m,n},$$

s.t.

(1) $\sum_{n=-N}^N p_{m,n} = \pi_m^X$ for $m \in [-N, N]$;
(2) $\sum_{m=-N}^N p_{m,n} = \pi_n^Y$ for $n \in [-N, N]$;
(3) $\sum_{m=\max(0,-j)-N}^{\min(0,-j)+N} u^m p_{m,m+j} = \pi_j^Z$ for $j \in [-N, N]$;
(4) $p_{m,n} \geq 0$ for all $m$ and $n$.

Assumption $[A1]$ implies that there exists a price density $P$ consistent with all option prices. So the solution set for LP1 is not empty. This program is bounded below by zero and above by $(\alpha x_0 u^N + \beta y_0 u^N - \gamma)^+$. Therefore, this program must have a solution.

Consider a strategy $\phi = (G, H, F)$ whose three components represent trading positions in currency options. Each component is represented by a $(2N+1)$ vector. To seek the cheapest dominating strategy, the hedging problem in (9) is reexpressed as follows

(LP2) find the $(2N+1) \times 1$ vectors $G$, $H$ and $F$ so that
$$\min_{G,H,F} \sum_m g_m \pi_m^X + \sum_n h_n \pi_n^Y + \sum_j f_j \pi_j^Z,$$

s.t.

(1') $g_m + h_n + u^m f_j \mathbf{1}_{j=n-m} \geq (\alpha x_0 u^m + \beta y_0 u^n - \gamma)^+$ for all $m, n, j$;
(2') $g_m, h_n, f_j \in \mathbb{R}$ for all $m, n, j$.

Since the program LP1 has a solution, the LP Duality Theorem implies that the program LP2 must also have a solution.

### 4.2. Numerical Results

To attain insights into the tightness of arbitrage bounds, we use a method of the trinomial tree to simulate the dynamics of the Euro ($X$) and Pound ($Y$) exchange rates against the U.S. dollar. It is assumed that (i) these two exchange rate processes are correlated with coefficient $\rho$ over time, and have an identical annual volatility, e.g., $\sigma^X = \sigma^Y = \sigma$; and (ii) both the domestic and foreign interest rate are zero.

For the sets in (22), the levels of the rate $X$ ($Y$) are bounded in a range of $[x_0 u^{-J}, x_0 u^J]$ ($[y_0 u^{-J}, y_0 u^J]$) where $0 < J < N$. In this way, there are absorbing boundaries imposed on rate levels for computational convenience. For a large $J$, these boundaries have no significant impact on numerical analysis.

All European option prices on $X$ and $Y$ are separately generated using the trinomial tree method over $N = 40$ time steps. The price densities $\pi^X$ and $\pi^Y$ are calculated from these European options. For a set of option prices $C(K)$, the price density $\pi$ is calculated as follows:

$$\pi(K) = \frac{C(Ku, t) - (1 + u)C(K) + uC(K/u)}{Ku - K},$$

for each $K$ in the set. By applying Boyle (1988)'s numerical procedure [28], we generate a (feasible) price density $P$ which is consistent with initial option prices on $X$ and $Y$. Finally, the price density $\pi^Z$ is constructed from the price density $P$. The density $\pi^Z$ is calculated by $\pi_j^Z = \sum_{m=\max(0,-j)-J}^{\min(0,-j)+J} u^m p_{m,m+j}$ for $j \in [-J, J]$.

All relevant parameters are set by

$$(x_0, y_0, T, \sigma, \rho, u, N, J) = (1.6, 2.5, 1, 0.42, 0.20, 1.12, 40, 30).$$

These parameters are employed with the following considerations:

- The exchange-rates of the Euro ($X$) and Pound ($Y$) against the U.S. dollar in the markets are higher than 1.0, and the former is relatively lower than the latter;
- The correlation parameter $\rho$ is positive, as both the Euro- and Pound-exchange rates against the U.S. dollar are usually positively correlated;
- Associated with the positive correlation between X and Y, it is assumed that they have the identical annualized volatility of 42% without the loss of generality.

As such, the left panel in Figure 3 illustrates the initial price density $P$ generated by Boyle's procedure. Each contour line represents the positive prices of this density function. The right panel in Figure 3 reports the price densities of Arrow–Debreu claims on three exchange rates.

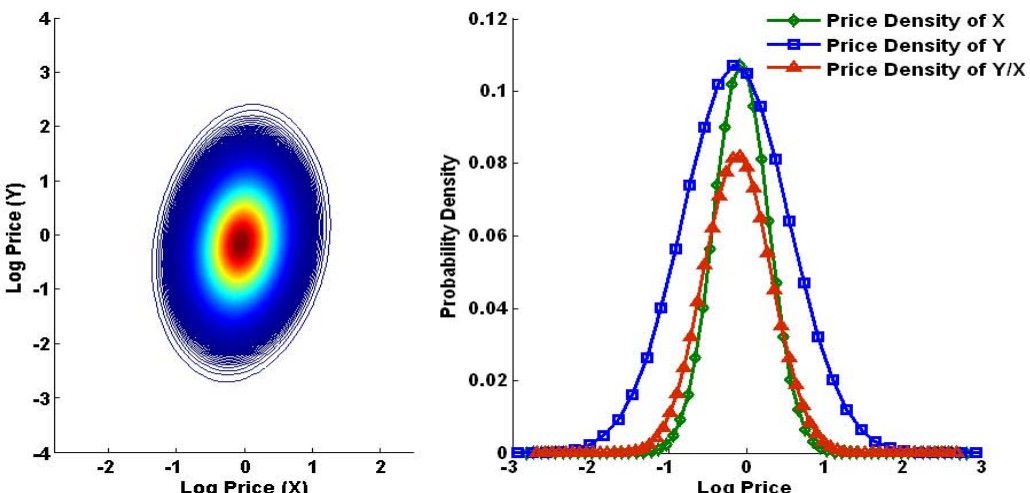

**Figure 3. Initial Probability Densities**. The time-zero prices of the exchange rates $X$ and $Y$ are $x_0 = 1.6$ and $y_0 = 2.5$. The volatilities of the processes $X$ and $Y$ are 42%. Both the domestic rate and foreign rate are zero. The marginal densities $\pi^X$ and $\pi^Y$ are attained within a price mesh of $u = 1.12$ and $J = 30$. The marginal density $\pi^Z$ is derived from the joint density that is calculated using the correlation $\rho = 0.20$. In the left panel, each contour line represents the positive prices of the initial joint density $P$.

The initial price density $P$ yields a price for a basket option. In order to measure the tightness of arbitrage bounds, this price can be viewed as a benchmark price, denoted as $V_{bs}$. Let $\overline{\mathcal{B}}_h$ and $\underline{\mathcal{B}}_h$ stand for the upper and lower valuation bound on this basket option, enforced by the option prices on $X$

and $Y$. Let $\overline{\mathcal{B}}_t$ and $\underline{\mathcal{B}}_t$ be valuation bounds if cross-currency options are traded as hedging instruments. These new instruments would tighten arbitrage bounds. To gauge the magnitude of the tightness, we consider the following measure:

$$\epsilon = 1 - \frac{\overline{\mathcal{B}}_t - \underline{\mathcal{B}}_t}{\overline{\mathcal{B}}_h - \underline{\mathcal{B}}_h}. \tag{23}$$

### 4.2.1. Valuation Bounds and Hedging Strategies

Now consider a basket option that pays $[1.2X + 0.9Y - 3.8]^+$ dollars after one year. Given the different sets of hedging instruments, the price functions that maximize this option's price have been characterized in Figure 1, while Figure 2 shows the price functions that minimize its price. Accordingly, the valuation bounds on this option are attained as follows:

$$(\underline{\mathcal{B}}_h, \underline{\mathcal{B}}_t, V_{bs}, \overline{\mathcal{B}}_t, \overline{\mathcal{B}}_h) = (0.38, 0.52, 0.72, 0.83, 0.86).$$

In terms of tightness, the bounds are substantially improved by $\epsilon = 36\%$ when information about the prices of cross-currency options is incorporated.

We further investigate the associated hedging strategies that enforce these valuation bounds. Figure 4 shows two dominating strategies that determine the upper bounds on the option **b**. First, the dominating strategy simply involves buying long 1.2 calls on $X$ with strike $K_a = 1.42$ and buying long 0.9 calls on $Y$ with strike $K_b = 2.23$. This strategy delivers the bound $\overline{\mathcal{B}}_h$. To attain the tight upper valuation bound $\overline{\mathcal{B}}_t$, the dominating strategy involves buying dollar-denominated puts and calls on $X$ and $Y$ and selling puts and calls on $Z$:

- buying an option portfolio on $X$ that consists of long 0.56 puts with strike 1.02, long 0.64, 0.56, 0.64 calls with strikes 1.28, 1.79, 2.25;
- buying an option portfolio on $Y$ that consists of long 0.57 puts with strike 1.59, long 0.33, 0.57, 0.33 calls with strikes 1.78, 2.80, 3.14;
- selling an option portfolio on $Z$ that consists of short 0.57 puts with strike 1.24 and short 0.33 calls with strike 1.75.

The left panel in Figure 4 reports the terminal payoff to this portfolio. As a result, this strategy is cheaper by about 0.04 dollars.

Accordingly, Figure 5 shows the dominated strategies that determine the lower bounds on the option **b**. The left panel illustrates the sub-replicating strategy that involves selling short 1.2 puts and buying long 1.2 calls on $X$ with strikes $K_a^1 = 1.42$ and $K_a^2 = 2.01$, and selling short 0.9 puts and buying long 0.9 calls on $Y$ with strikes $K_b^1 = 1.99$ and $K_b^2 = 2.80$. The right panel presents the second sub-replicating strategy that leads to the tighter lower valuation bound. This dominated strategy can be described as follows:

- buying an option portfolio on $X$ that consists of long 1.52 puts with strike 1.72 short 0.32 calls with strike 2.24;
- buying an option portfolio on $Y$ that consists of long 1.44 puts with strike 2.23, short 0.54 calls with strike 3.93;
- selling a portfolio of the cross-currency options on $Z$ that consists of short 0.90 puts with strike 0.88 and short 0.54 calls with strike 2.45.

Equivalently, this strategy improves the lower bound by about 0.14 dollars.

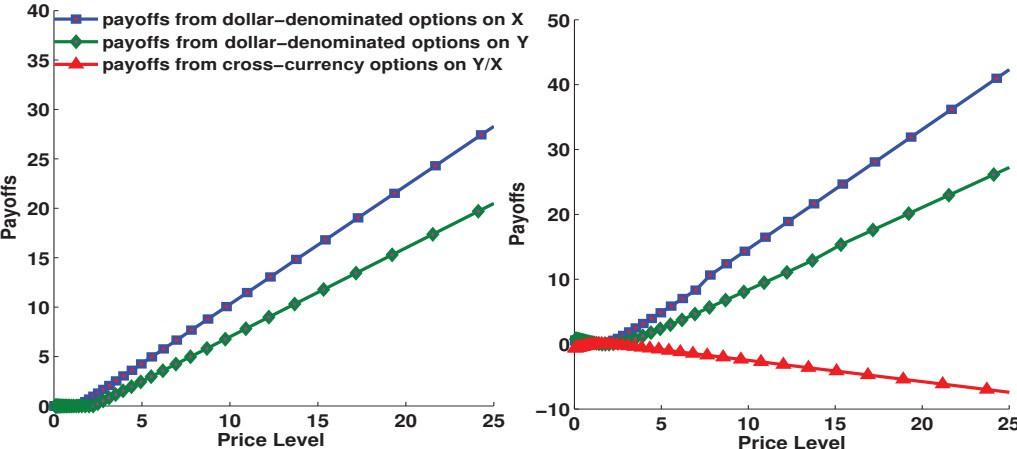

**Figure 4. Terminal Payoffs to Dominating Strategies**. The time-zero prices of the exchange rates $X$ and $Y$ are 1.6 and 2.5. The volatilities of the processes $X$ and $Y$ are 42%, and two processes are correlated with $\rho = 0.20$. Both the domestic and foreign interest rate are zero. The basket option with 1-year maturity has the payoff $[1.2X + 0.9Y - 3.8]^+$. The dominating strategies are attained within a price mesh of $u = 1.12$ and $J = 30$. In the left panel, the relevant strike on $X$ ($Y$) is 1.42 (2.23). In the right panel, the relevant strikes on the $X$ ($Y$) are $(1.02, 1.28, 1.79, 2.25)$ $((1.59, 1.78, 2.80, 3.14))$, and the strikes on $Z$ are $(1.24, 1.75)$.

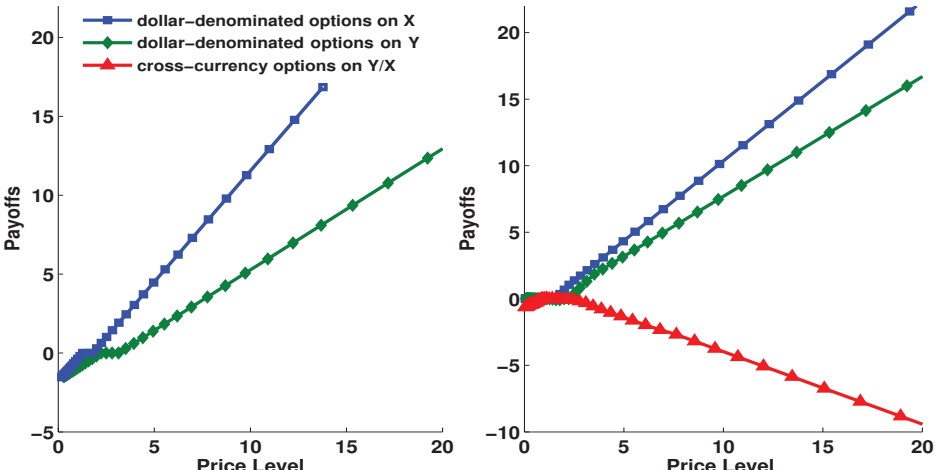

**Figure 5. Static Sub-replicating Strategies**. The time-zero prices of the exchange rates $X$ and $Y$ are 1.6 and 2.5. The volatilities of the processes $X$ and $Y$ are 42%, and two processes are correlated with $\rho = 0.20$. Both the domestic and foreign interest rate are zero. The basket option **b** with 1-year maturity has the payoff $[1.2X + 0.9Y - 3.8]^+$. The dominated strategies are attained within a price mesh of $u = 1.12$ and $J = 30$. In the left panel, the relevant strike on $X$ ($Y$) is $(1.42, 2.01)$ $((1.99, 2.80))$. In the right panel, the relevant strikes on $X$ ($Y$) are $(1.72, 2.24)$ $((2.23, 3.93))$, and the strikes on $Z$ are $(0.88, 2.45)$.

### 4.2.2. Sensitivity of Valuation Bounds

As shown in Figure 3, an appropriate $J$ is chosen so that three price densities are close to zero when exchange rate levels reach the absorbing boundaries. In this way, valuation bounds on basket options are relatively independent of the number of price levels ($N$). We now investigate the sensitivity of valuation bounds by varying the jump size ($u$), as reported in Table 1. In terms of the measure

in (23), all the numbers indicate that the price bounds on two options are significantly tightened by incorporating price information about cross-currency options. These numbers also show that these valuation bounds are relatively robust to changes in the jump size $u$. The price bounds on the first (second) option can at least be improved by an average of 36% (32%)

**Table 1.** **Sensitivity of Valuation Bounds Against Jump Size.** The time-zero prices of the exchange rates $X$ and $Y$ are 1.6 and 2.5. The volatilities of the processes $X$ and $Y$ are 42%, and two processes are correlated with $\rho = 0.20$. Both the domestic and foreign interest rate are zero. All prices are attained within a price mesh of $J = 30$. Two basket options mature at $T = 1$ (year).

| u | $\underline{\mathcal{B}}_h$ | $\underline{\mathcal{B}}_t$ | $V_{bs}$ | $\overline{\mathcal{B}}_t$ | $\overline{\mathcal{B}}_h$ | $\epsilon$ | $\underline{\mathcal{B}}_h$ | $\underline{\mathcal{B}}_t$ | $V_{bs}$ | $\overline{\mathcal{B}}_t$ | $\overline{\mathcal{B}}_h$ | $\epsilon$ |
|---|---|---|---|---|---|---|---|---|---|---|---|---|
| | | | $[1.2X + 0.9Y - 3.8]^+$ | | | | | | $[-X - Y + 4.8]^+$ | | | |
| 1.12 | 0.386 | 0.527 | 0.718 | 0.832 | 0.863 | 36.0% | 0.782 | 0.874 | 0.994 | 1.117 | 1.143 | 32.2% |
| 1.14 | 0.389 | 0.534 | 0.717 | 0.825 | 0.862 | 38.6% | 0.783 | 0.875 | 0.993 | 1.115 | 1.139 | 32.5% |
| 1.16 | 0.388 | 0.538 | 0.714 | 0.825 | 0.863 | 39.5% | 0.780 | 0.873 | 0.991 | 1.111 | 1.134 | 32.6% |
| 1.18 | 0.390 | 0.583 | 0.712 | 0.818 | 0.863 | 50.4% | 0.783 | 0.871 | 0.987 | 1.106 | 1.131 | 32.4% |
| 1.20 | 0.371 | 0.577 | 0.709 | 0.815 | 0.865 | 51.8% | 0.784 | 0.876 | 0.985 | 1.105 | 1.134 | 34.4% |

In order to investigate the sensitivity of valuation bounds to changes in correlation between the Euro and Pound exchange rates, we assume that the coefficient $\rho$ varies in the range of $[-1, 1]$. The arbitrage bound $\mathcal{B}_h$ is independent of correlation coefficient, as this bound is enforced by a portfolio of dollar-denominated options. Nevertheless, changes in correlation coefficient have impact on the valuation bound $\mathcal{B}_t$, as this bound depends on the density $\pi^Z$ that is constructed from an initial price density $P$. For each $\rho$, we use Boyle's procedure [28] to generate a new price density $P$.

Table 2 presents the sensitivity of valuation bounds when correlation coefficient varies. The bounds $\mathcal{B}_t$ on two options increase as the coefficient $\rho$ increases. In particular, the bounds $\overline{\mathcal{B}}_t$ approach $\mathcal{B}_h$ when the coefficient $\rho$ increases to 1. However, the tightness of valuation bounds decreases as the the coefficient $\rho$ increases from $-0.8$ to 0 or decreases from 0.8 to 0. For $\rho = -0.8$, the valuation bounds on the first (second) option is substantially tightened by about 73.3% (81.2%). Similarly, the bounds on these two options are also greatly tightened for $\rho = 0.8$, compared with the tightness of bounds at $\rho = 0$ (35.0% and 35.5% respectively).

**Table 2.** **Sensitivity of Valuation Bounds Against Correlation Coefficient.** The time-zero prices of the exchange rates $X$ and $Y$ are 1.6 and 2.5. The volatilities of the processes $X$ and $Y$ are 42%. Both the domestic and foreign interest rate are zero. All prices are attained within a price mesh of $u = 1.12$ and $J = 30$. Two basket options mature at $T = 1$ (year).

| $\rho$ | $\underline{\mathcal{B}}_h$ | $\underline{\mathcal{B}}_t$ | $V_{bs}$ | $\overline{\mathcal{B}}_t$ | $\overline{\mathcal{B}}_h$ | $\epsilon$ | $\underline{\mathcal{B}}_h$ | $\underline{\mathcal{B}}_t$ | $V_{bs}$ | $\overline{\mathcal{B}}_t$ | $\overline{\mathcal{B}}_h$ | $\epsilon$ |
|---|---|---|---|---|---|---|---|---|---|---|---|---|
| | | | $[1.2X + 0.9Y - 3.8]^+$ | | | | | | $[-X - Y + 4.8]^+$ | | | |
| $-0.80$ | 0.387 | 0.392 | 0.483 | 0.519 | 0.862 | 73.3% | 0.783 | 0.793 | 0.805 | 0.860 | 1.142 | 81.2% |
| $-0.60$ | 0.386 | 0.401 | 0.545 | 0.608 | 0.861 | 56.4% | 0.783 | 0.802 | 0.845 | 0.924 | 1.142 | 66.1% |
| $-0.40$ | 0.386 | 0.425 | 0.597 | 0.678 | 0.860 | 46.5% | 0.783 | 0.815 | 0.887 | 0.981 | 1.142 | 53.7% |
| $-0.20$ | 0.386 | 0.453 | 0.644 | 0.735 | 0.861 | 40.5% | 0.783 | 0.833 | 0.927 | 1.035 | 1.142 | 44.0% |
| 0 | 0.380 | 0.500 | 0.706 | 0.817 | 0.860 | 35.0% | 0.782 | 0.852 | 0.966 | 1.084 | 1.142 | 35.5% |
| 0.20 | 0.388 | 0.530 | 0.717 | 0.821 | 0.860 | 38.3% | 0.782 | 0.874 | 0.994 | 1.116 | 1.142 | 32.7% |
| 0.40 | 0.388 | 0.584 | 0.757 | 0.859 | 0.860 | 41.7% | 0.782 | 0.910 | 1.033 | 1.142 | 1.142 | 35.7% |
| 0.60 | 0.383 | 0.652 | 0.793 | 0.860 | 0.860 | 55.4% | 0.782 | 0.961 | 1.069 | 1.142 | 1.142 | 49.9% |
| 0.80 | 0.388 | 0.741 | 0.829 | 0.860 | 0.860 | 74.5% | 0.782 | 1.023 | 1.105 | 1.142 | 1.142 | 67.2% |

Since the dependence between the Euro ($X$) and Pound ($Y$) has impact on the joint price density $P$, the construction of the density $\pi^Z$ means that increasing (decreasing) $\rho$ leads to increases (decreases) in the kurtosis of the density $\pi^Z$. For a large $\rho$ (close to 1), the density $\pi^Z$ with high kurtosis implies that cross-currency options have limited impact on the tightness of upper bounds but significantly improve lower bounds, as shown in Table 2. This impact on both lower and upper price bounds may be substantial when the coefficient $\rho$ decreases to $-1$, as there are more Arrow–Debreu securities with non-zero prices for trading.

We further examine the sensitivity of price bounds against other parameters (i.e., maturity, volatility and non-zero yield rate). Table 3 reports the sensitivity of valuation bounds to different maturities. In term of magnitude, the price bounds on the first (second) option are increasingly tightened from 32.3% (25.3%) to 39.6% (35.1%) as maturity increases from 6 months to 15 months. The similar result is observed in Table 4. The price bounds on the first (second) option are increasingly improved from 29.3% (21.5%) to 38.5% (33.5%) as volatility level increases by 80%.

**Table 3. Sensitivity of Valuation Bounds Against Maturity.** The time-zero prices of the exchange rates $X$ and $Y$ are 1.6 and 2.5. The volatilities of the processes $X$ and $Y$ are 42%, and two processes are correlated with $\rho = 0.20$. Both the domestic and foreign interest rate are zero. All prices are attained within a price mesh of $u = 1.12$ and $J = 30$.

| T (Year) | $[1.2X + 0.9Y - 3.8]^+$ | | | | | | $[-X - Y + 4.8]^+$ | | | | | |
| | $\underline{\mathcal{B}}_h$ | $\underline{\mathcal{B}}_t$ | $V_{bs}$ | $\overline{\mathcal{B}}_t$ | $\overline{\mathcal{B}}_h$ | $\epsilon$ | $\underline{\mathcal{B}}_h$ | $\underline{\mathcal{B}}_t$ | $V_{bs}$ | $\overline{\mathcal{B}}_t$ | $\overline{\mathcal{B}}_h$ | $\epsilon$ |
|---|---|---|---|---|---|---|---|---|---|---|---|---|
| 0.50 | 0.370 | 0.449 | 0.575 | 0.655 | 0.674 | 32.3% | 0.725 | 0.772 | 0.848 | 0.938 | 0.947 | 25.3% |
| 0.75 | 0.376 | 0.484 | 0.651 | 0.751 | 0.775 | 33.0% | 0.751 | 0.823 | 0.925 | 1.035 | 1.052 | 29.7% |
| 1.00 | 0.388 | 0.531 | 0.717 | 0.821 | 0.860 | 38.3% | 0.782 | 0.874 | 0.994 | 1.116 | 1.114 | 32.7% |
| 1.25 | 0.391 | 0.571 | 0.777 | 0.898 | 0.934 | 39.6% | 0.815 | 0.921 | 1.057 | 1.191 | 1.222 | 33.9% |
| 1.50 | 0.426 | 0.615 | 0.831 | 0.965 | 1.006 | 39.6% | 0.849 | 0.967 | 1.145 | 1.257 | 1.295 | 35.1% |

**Table 4. Sensitivity of Valuation Bounds Against Volatility.** The time-zero prices of the exchange rates $X$ and $Y$ are 1.6 and 2.5. It is assumed that the underlying processes $X$ and $Y$ are correlated with $\rho = 0.20$ with the identical annualized volatility, e.g., $\sigma^X = \sigma^Y = \sigma$. Both the domestic and foreign interest rate are zero. All prices are attained within a price mesh of $u = 1.12$ and $J = 30$. Two basket options mature at $T = 1$ (year).

| $\sigma$ | $[1.2X + 0.9Y - 3.8]^+$ | | | | | | $[-X - Y + 4.8]^+$ | | | | | |
| | $\underline{\mathcal{B}}_h$ | $\underline{\mathcal{B}}_t$ | $V_{bs}$ | $\overline{\mathcal{B}}_t$ | $\overline{\mathcal{B}}_h$ | $\epsilon$ | $\underline{\mathcal{B}}_h$ | $\underline{\mathcal{B}}_t$ | $V_{bs}$ | $\overline{\mathcal{B}}_t$ | $\overline{\mathcal{B}}_h$ | $\epsilon$ |
|---|---|---|---|---|---|---|---|---|---|---|---|---|
| 25% | 0.371 | 0.427 | 0.523 | 0.592 | 0.606 | 29.3% | 0.712 | 0.743 | 0.799 | 0.873 | 0.878 | 21.5% |
| 30% | 0.370 | 0.451 | 0.578 | 0.659 | 0.679 | 32.5% | 0.726 | 0.774 | 0.852 | 0.942 | 0.952 | 25.6% |
| 35% | 0.372 | 0.481 | 0.635 | 0.732 | 0.754 | 34.3% | 0.745 | 0.812 | 0.909 | 1.012 | 1.030 | 29.9% |
| 40% | 0.384 | 0.511 | 0.694 | 0.801 | 0.829 | 35.1% | 0.771 | 0.855 | 0.969 | 1.088 | 1.109 | 31.3% |
| 45% | 0.391 | 0.551 | 0.754 | 0.869 | 0.908 | 38.5% | 0.802 | 0.902 | 1.032 | 1.161 | 1.190 | 33.5% |

Table 5 shows that changing interest rates has different impact on two basket options. The tightness of the price bounds on the first option ($\alpha, \beta, \gamma > 0$) is decreasing as the foreign interest rate $r_f$ decreases, while the bounds on the second option ($\alpha, \beta, \gamma < 0$) are increasingly improved. Overall, the valuation bounds on two options can be tightened by cross-currency options for different interest rate levels. The tightness of the price bounds on the first (second) option has an average of about 31.7% (31.5%) when the domestic interest rate is set as 2% and the foreign interest rate varies in a range of $[-8\%, 12\%]$.

**Table 5. Sensitivity of Valuation Bounds Against Yield Rate.** The time-zero prices of the exchange rates $X$ and $Y$ are 1.6 and 2.5. The volatilities of the processes $X$ and $Y$ are 42%, and two processes are correlated with $\rho = 0.20$. The yield rate ($\delta$) is the difference between the domestic risk-free interest rate ($r_d^\$$) and foreign interest rate ($r_f$), i.e., $\delta = r_d^\$ - r_f$. The domestic interest rate is set by 2%, and the foreign interest rates for the Euro ($X$) and Pound ($Y$) are assumed to be identical. All prices are attained within a price mesh of $u = 1.12$ and $J = 30$. Two basket options mature at $T = 1$ (year), and thus the discount factor is expressed as $DF = e^{-r_d^\$ T} = 0.98$.

| Yield Rate | $[1.2X + 0.9Y - 3.8]^+$ | | | | | | $[-X - Y + 4.8]^+$ | | | | | |
| | $\underline{\mathcal{B}}_h$ | $\underline{\mathcal{B}}_t$ | $V_{bs}$ | $\overline{\mathcal{B}}_t$ | $\overline{\mathcal{B}}_h$ | $\epsilon$ | $\underline{\mathcal{B}}_h$ | $\underline{\mathcal{B}}_t$ | $V_{bs}$ | $\overline{\mathcal{B}}_t$ | $\overline{\mathcal{B}}_h$ | $\epsilon$ |
|---|---|---|---|---|---|---|---|---|---|---|---|---|
| $-10\%$ | 0.157 | 0.313 | 0.464 | 0.564 | 0.598 | 43.1% | 1.108 | 1.153 | 1.234 | 1.348 | 1.353 | 20.4% |
| $-5\%$ | 0.237 | 0.405 | 0.575 | 0.680 | 0.714 | 42.1% | 0.938 | 1.003 | 1.103 | 1.222 | 1.234 | 26.7% |
| 0% | 0.380 | 0.520 | 0.704 | 0.805 | 0.843 | 38.3% | 0.766 | 0.856 | 0.974 | 1.094 | 1.119 | 32.7% |
| $+5\%$ | 0.575 | 0.656 | 0.849 | 0.969 | 0.987 | 23.9% | 0.596 | 0.713 | 0.851 | 0.973 | 1.007 | 36.9% |
| $+10\%$ | 0.795 | 0.819 | 1.015 | 1.135 | 1.151 | 11.0% | 0.429 | 0.581 | 0.735 | 0.857 | 0.901 | 41.3% |

## 5. Conclusions

Basket options are traded as alternative instruments to manage risk exposure in multiple underlying assets. Despite their attractive features, pricing basket options poses a challenge to

practitioners. Through this paper, we have studied the problem of valuing currency basket options. Instead of precise prices, we derive model-free valuation bounds on these options.

We identify a collection of tradable options on individual currency pairs as hedging instruments. Three currencies, the Euro, British pound and U.S. dollar are considered. We emphasize the role of price information about cross-currency options in pricing basket options. First of all, arbitrage bounds are derived from portfolios that involve only dollar-denominated options. If cross-currency options are additionally traded, price bounds on basket options can be further tightened. These bounds are enforced by static portfolios of both dollar-denominated options and cross-currency options.

We have derived both upper and lower bounds on currency basket options. These valuation bounds are enforced by static hedging strategies that are constructed from available hedging instruments, e.g., the options on individual currency pairs. Meanwhile, these strategies are associated with the joint densities of underlying currencies that can be characterized.

**Funding:** This work was supported by Ministry of Education (MoE) (15YJC790026), Xi'an Jiaotong-Liverpool University (RDF-14-02-51) and XJTLU Key Program Special Fund (KSF-P-02 and KSF-E-31).

**Acknowledgments:** The author would like to thank the referees for carefully reading my manuscript and for giving such constructive comments which substantially helped improving the quality of my paper. Moreover, the author wish to thank the Editor for helpful cooperation that have improved the layout of the paper. The author appreciates the financial support from the Young Scholar Programme by Ministry of Education (MoE) (15YJC790026), Xi'an Jiaotong-Liverpool University (RDF-14-02-51) and XJTLU Key Program Special Fund (KSF-P-02 and KSF-E-31).

**Conflicts of Interest:** The author declares no conflict of interest regarding the publication of the research article.

## Appendix A. Proof of Lemma 1

**Proof.** Let $C(K)$ be the market price of an option with payoff $(S - K)^+$ for $K \in \mathbb{R}^+$. Since a continuum of option prices is available at inception and these prices are twice differentiable, the price density of Arrow–Debreu claims is known from Breeden and Litzenberger (1978) [25]:

$$\pi(k) = \frac{\partial^2 C(K)}{\partial K^2}\Big|_{K=k}, \tag{A1}$$

for each strike $k$. Given the prices of the dollar-denominated options on $X$ and $Y$, and the $X$-denominated options on $Y$, there exists a price density for each currency pair.

Let $p(x, y)$ represent the price density of a claim that pays \$1 if $X = x$ and $Y = y$. The non-arbitrage argument in assumption [A1] implies that there must exist a risk-neutral probability measure $\mu$ such that

$$\pi^X(x) = \mathbb{E}^\mu[1_{X=x}] = \mathbb{E}^\mu[\int_{\mathbb{R}^+} 1_{X=x} 1_{Y=y} dy] = \int_{\mathbb{R}^+} \mathbb{E}^\mu[1_{X=x} 1_{Y=y}] dy = \int_{\mathbb{R}^+} p(x, y) dy, \tag{A2}$$

given the dollar as the common currency. Similarly, we also have

$$\pi^Y(y) = \mathbb{E}^\mu[1_{Y=y}] = \int_{\mathbb{R}^+} p(x, y) dx. \tag{A3}$$

Since the $X$-denominated price of a claim that pays 1 unit $X$ if $X = x$ and $Y = xz$ has the density $\pi^Z(z)$, it is equivalent to stating that

- the time-0 price density of this claim is $x_0 \pi^Z(z)$;
- the time-$T$ payoff of this claim is $\int_{\mathbb{R}^+} x 1_{X=x} 1_{Y=xz} dx$,

if the U.S. dollar is the base currency. This statement leads to the third equality

$$\pi^Z(z) = \frac{1}{x_0} \mathbb{E}^\mu[\int_{\mathbb{R}^+} x 1_{X=x} 1_{Y=xz} dx] = \frac{1}{x_0} \int_{\mathbb{R}^+} x p(x, xz) dx. \tag{A4}$$

$\square$

## Appendix B. Proof of Lemma 2

**Proof.** Given the currency $X$ as the base, let the function $p_X(x, y)$ be the price density of a claim that pays 1 unit $X$ if $1/X = x$ and $Y/X = y$. Equivalently, this claim pays $1/x$ dollars, and its dollar-denominated price has a density $x_0 p_X(x, y)$. Recall that the function $p(1/x, y/x)$ is the dollar-denominated price density of a claim that pays one dollar if $X = 1/x$ and $Y = y/x$. To avoid arbitrage, we then have

$$x_0 p_X(x, y) = \frac{1}{x} p\left(\frac{1}{x}, \frac{y}{x}\right), \tag{A5}$$

which immediately results in the first equality

$$\int_{\mathbb{R}^+} p_X(x, y) dy = \int_{\mathbb{R}^+} \frac{1}{x_0 x} p\left(\frac{1}{x}, \frac{y}{x}\right) d\left(\frac{y}{x}\right) = \frac{1}{x_0 x} \pi^X\left(\frac{1}{x}\right). \tag{A6}$$

This equality states that the dollar-denominated price density of a claim that pays 1 unit $X$ if $1/X = x$ is equal to $\pi^X(1/x)/(x_0 x)$.

Also, it is easy to derive the following result:

$$\int_{\mathbb{R}^+} p_X(x, y) dx = \pi^Z(y), \tag{A7}$$

which determines the $X$-denominated price density of a claim that pays 1 unit $X$ if $Y/X = y$.

To derive the third equality, consider a claim that pays 1 unit $X$ if $1/X = x$ and $Y/X = xz$ for $z \in \mathbb{R}^+$. Its $X$-denominated price density is $p_X(x, xz)$. Meanwhile, it is known that the dollar-denominated price density of a claim that pays \$1 if $Y = z$ is $\pi^Y(z)$ such that

$$x_0 \int_{\mathbb{R}^+} x p_X(x, xz) dx = \int_{\mathbb{R}^+} p\left(\frac{1}{x}, z\right) d\left(\frac{1}{x}\right) = \pi^Y(z), \tag{A8}$$

which yields the third equality. $\square$

## Appendix C. Proof of Proposition 2

**Proof.** Given the dollar as the currency base, suppose that there exists a pair of $(p, \phi)$ ($p \in \mathcal{P}, \phi \in \mathcal{A}$) that solves two programs in (7) and (9). From Lemma 2, a price density in the currency base $X$ can be determined as follows:

$$p_X(x, y) = \frac{1}{x_0 x} p\left(\frac{1}{x}, \frac{y}{x}\right), \text{ for } (x, y) \in \mathbb{R}_2^+. \tag{A9}$$

Then if the function $p$ maximizes the dollar-denominated price of a basket option, the new function $p_X$ also maximizes the $X$-denominated price of this option in order to avoid arbitrage.

To see that the strategy $\phi$ dominates the basket option based on the currency $X$, consider its $X$-denominated payoff:

$$[\alpha + \beta y - \gamma x]^+, \text{ for } (x, y) \in \mathbb{R}_2^+. \tag{A10}$$

As the dual of pricing, hedging this payoff would involve a portfolio of both the dollar-denominated options on $X$ and $Y$ and the $X$-denominated options on $Y$:

$$g\left(\frac{1}{x}\right)x + xh(w)1_{w=y/x} + f(y) \geq [\alpha + \beta y - \gamma x]^+, \tag{A11}$$

which is equivalent to holding $g\left(\frac{1}{x}\right)$ claims if $1/X = x$, $h(w)$ claims if $Y = w$ and $f(y)$ claims if $Y/X = y$ at inception. This inequality may be expressed as in dollars

$$g\left(\frac{1}{x}\right) + h(w) + \frac{1}{x} f(y)1_{y=xw} \geq \left[\alpha \frac{1}{x} + \beta \frac{y}{x} - \gamma\right]^+. \tag{A12}$$

This is identical to the expression by setting $x = \frac{1}{\hat{x}}$

$$g(\hat{x}) + h(w) + \hat{x}f(y)1_{y=w/\hat{x}} \geq [\alpha\hat{x} + \beta w - \gamma]^+. \tag{A13}$$

Since the strategy $\phi = (g, h, f)$ solves the dual problem in (9), the inequality above implies that this strategy also dominates the basket option in the currency base $X$. $\square$

**Appendix D. Proof of Proposition 3**

**Proof.** To derive upper bounds, consider the dollar-denominated payoff of a basket option in (4):

$$(\alpha X + \beta Y - \gamma)^+ \leq \alpha(X - \frac{\gamma\lambda_1}{\alpha})^+ + \beta(Y - \frac{\gamma\lambda_2}{\beta})^+, \text{ for } (\alpha, \beta, \gamma) \in \mathbb{R}_3^+, \tag{A14}$$

for $\lambda_1, \lambda_2 \geq 0$ and $\lambda_1 + \lambda_2 = 1$.

By setting $K_a = \frac{\gamma\lambda_1}{\alpha} \geq 0$ and $K_b = \frac{\gamma\lambda_2}{\beta} \geq 0$, the upper bound on this basket option is determined as follows:

$$\overline{\mathcal{B}} = \min_{K_a, K_b \geq 0: \alpha K_a + \beta K_b = \gamma} \alpha \int_{\mathbb{R}^+} (x - K_a)^+ \pi^X(x)dx + \beta \int_{\mathbb{R}^+} (y - K_b)^+ \pi^Y(y)dy. \tag{A15}$$

Two strikes $K_a$ and $K_b$ are determined by $\lambda_1$ and $\lambda_2$. The solution set for this program is not empty. This program is bounded above, and so it must have a solution. Let $K_a^*$ and $K_b^*$ be the optimal solutions for this program such that

$$\overline{\mathcal{B}} = \alpha \int_{\mathbb{R}^+} (x - K_a^*)^+ \pi^X(x)dx + \beta \int_{\mathbb{R}^+} (y - K_b^*)^+ \pi^Y(y)dy. \tag{A16}$$

To see that there exists a density function $p$ that supports this bound, define two separated sets

$$\begin{aligned} A &= (0, K_a^*] \times (0, K_b^*] \cup (K_a^*, \infty) \times (K_b^*, \infty); \\ B &= \mathbb{R}_2^+ \backslash A. \end{aligned} \tag{A17}$$

Consider a candidate density function:

$$p^*(x, y) = \begin{cases} \geq 0, & \text{if } (x, y) \in A; \\ 0, & \text{if } (x, y) \in B, \end{cases} \tag{A18}$$

such that $\int_{\mathbb{R}^+} p^*(x, y)dy = \pi^X(x)$ and $\int_{\mathbb{R}^+} p^*(x, y)dx = \pi^Y(y)$.

To ensure the existence of such a density function, consider the construction of a process $(X, Y)$ so that the variable $Y$ is an increasing function of $X$. Given any $v \sim U[0, 1]$, there exists a real number vector $(\bar{x}, \bar{y}) \in \mathbb{R}_2^+$ and the random variable vector $(X, Y)$ so that

$$prob(X \leq \bar{x}) = prob(Y \leq \bar{y}) = v. \tag{A19}$$

A bivariate process $(X, Y)$ is constructed through the inverse function, and so this process is comonotonic. This process implies that:

(i) the events $(X \leq \bar{x})$ and $(Y \leq \bar{y})$ are mutually compatible each other and so do the events $(X > \bar{x})$ and $(Y > \bar{y})$;
(ii) the events $(X > \bar{x})$ and $(Y \leq \bar{y})$ are mutually exclusive each other and so do the events $(X \leq \bar{x})$ and $(Y > \bar{y})$.

By setting $\bar{x} = K_a^*$ and $\bar{y} = K_b^*$, we have the following equality in the regions where the events $(X > \bar{x}, Y > \bar{y})$ or $(X \le \bar{x}, Y \le \bar{y})$ occur:

$$(\alpha X + \beta Y - \gamma)^+ = \alpha(X - K_a^*)^+ + \beta(Y - K_b^*)^+. \tag{A20}$$

In the regions where the events $(X \le \bar{x}, Y > \bar{y})$ or $(X > \bar{x}, Y \le \bar{y})$ occur, we have the following inequality:

$$(\alpha X + \beta Y - \gamma)^+ < \alpha(X - K_a^*)^+ + \beta(Y - K_b^*)^+. \tag{A21}$$

For the candidate density function $p^*$, the price of the basket option can be expressed as follows:

$$\mathbb{E}_{p^*}[(\alpha x + \beta y - \gamma)^+] = \alpha \int_{\mathbb{R}^+} (x - K_a^*)^+ \pi^X(x)dx + \beta \int_{\mathbb{R}^+} (y - K_b^*)^+ \pi^Y(y)dy = \overline{\mathcal{B}}, \tag{A22}$$

which completes the proof. $\square$

**Appendix E. Proof of Proposition 4**

**Proof.** (1) We first set up a class of dominating hedging portfolios. Eight price levels are chosen from $X$ and $Y$

$$\begin{aligned} 0 < K_1 < K_2 < K_3 < K_4; \\ 0 < K_1^b < K_2^b < K_3^b < K_4^b, \end{aligned} \tag{A23}$$

so that these prices determine two price levels on $Z$:

$$z_1 = \frac{K_1^b}{K_2} = \frac{K_3^b}{K_4}, z_2 = \frac{K_2^b}{K_1} = \frac{K_4^b}{K_3}. \tag{A24}$$

To dominate the payoff of a basket option in (4), consider the strategy $\phi$ which consists of three components:

(i) the holdings of the dollar-denominated options on $X$ are

- long $(\alpha - \lambda)$ puts at strike $K_1$;
- long $\lambda$ calls at strike $K_2$;
- long $(\alpha - \lambda)$ calls at strike $K_3$;
- long $\lambda$ calls at strike $K_4$;

(ii) the holdings of the dollar-denominated options on $Y$ are

- long $(\beta - \delta)$ puts at strike $K_1^b$;
- long $\delta$ calls at strike $K_2^b$;
- long $(\beta - \delta)$ calls at strike $K_3^b$;
- long $\delta$ calls at strike $K_4^b$;

(iii) the holdings of the $X$-denominated options on $Y$ are

- short $(\beta - \delta)$ puts at strike $z_1$;
- short $\delta$ calls at strike $z_2$.

This strategy would lead to the terminal payoff

$$g(x) = \begin{cases} (\alpha - \lambda)(K_1 - x)^+, & \text{if } x \leq K_1; \\ 0, & \text{if } K_1 < x < K_2; \\ \lambda(x - K_2)^+, & \text{if } K_2 \leq x < K_3; \\ \alpha(x - K_3)^+ + \lambda(K_3 - K_2) & \text{if } K_3 \leq x < K_4; \\ \lambda(x - K_4)^+ + \alpha(x - K_3)^+ + \lambda(K_3 - K_2) & \text{if } K_4 \leq x; \end{cases}$$

$$h(y) = \begin{cases} (\beta - \delta)(K_1^b - y)^+, & \text{if } y \leq K_1^b; \\ 0, & \text{if } K_1^b < y < K_2^b; \\ \delta(y - K_2^b)^+, & \text{if } K_2^b \leq y < K_3^b; \\ \beta(y - K_3^b)^+ + \delta(K_3^b - K_2^b) & \text{if } K_3^b \leq y < K_4^b; \\ \delta(y - K_4^b)^+ + \beta(y - K_3^b)^+ + \delta(K_3^b - K_2^b) & \text{if } K_4^b \leq y; \end{cases} \tag{A25}$$

$$f(z) = \begin{cases} -(\beta - \delta)(z_1 - z)^+, & \text{if } 0 < z \leq z_1; \\ 0, & \text{if } z_1 < z < z_2; \\ -\delta(z - z_2)^+, & \text{if } z_2 \leq z. \end{cases}$$

From the program (12), there are three equalities:

$$(\beta - \delta)z_1 = \lambda; \delta z_2 = \alpha - \lambda, \text{ for } 0 < z_1 \leq \tfrac{\alpha}{\beta} \leq z_2;$$

$$\lambda(K_3 - K_2) + \delta(K_3^b - K_2^b) = \alpha K_3 + \beta K_3^b - \gamma (> 0), \tag{A26}$$

for $\lambda \in (0, \alpha)$ and $\delta \in (0, \beta)$. The equalities in (A24) equivalently state that

$$z_1 = \frac{K_3^b - K_1^b}{K_4 - K_2}, z_2 = \frac{K_4^b - K_2^b}{K_3 - K_1}. \tag{A27}$$

From these equalities, we can derive the following (in)equalities:

$$\lambda(K_1 - K_2) + \delta(K_3^b - K_4^b) = \alpha K_1 + \beta K_3^b - \gamma \leq 0;$$

$$\lambda(K_3 - K_4) + \delta(K_1^b - K_2^b) = \alpha K_3 + \beta K_1^b - \gamma \leq 0; \tag{A28}$$

$$\alpha K_4 + \beta K_2^b \geq \gamma; \alpha K_2 + \beta K_4^b \geq \gamma.$$

These (in)equalities can establish the following results:

$$g(x) + h(y) + xf(z)1_{z=y/x} - [\alpha x + \beta y - \gamma]^+ = \begin{cases} 0, \text{ if } 0 < x \leq K_1 \text{ and } K_2^b \leq y \leq K_3^b; \\ 0, \text{ if } K_1 \leq x \leq K_2 \text{ and } K_1^b \leq y \leq K_2^b; \\ 0, \text{ if } K_2 \leq x \leq K_3 \text{ and } 0 < y \leq K_1^b; \\ 0, \text{ if } K_2 \leq x \leq K_3 \text{ and } y \geq K_4^b; \\ 0, \text{ if } K_3 \leq x \leq K_4 \text{ and } K_3^b \leq y \leq K_4^b; \\ 0, \text{ if } x \geq K_4 \text{ and } K_2^b \leq y \leq K_3^b; \\ > 0, \text{ otherwise.} \end{cases} \tag{A29}$$

Therefore, the strategy $\phi$ dominates the basket option in (4). As a result, the minimum cost of such a strategy would provide an upper price bound for this basket option:

$$\min_{\lambda, \delta, z_1, z_2, K_i, i=1,2,3,4} \int_{\mathbb{R}^+} [(\alpha - \lambda)(K_1 - x)^+ + \lambda(x - K_2)^+$$

$$+ (\alpha - \lambda)(x - K_3)^+ + \lambda(x - K_4)^+]\pi^X(x)dx$$

$$+ \int_{\mathbb{R}^+} [(\beta - \delta)(K_1^b - y)^+ + \delta(y - K_2^b)^+ \tag{A30}$$

$$+ (\beta - \delta)(y - K_3^b)^+ + \delta(y - K_4^b)^+]\pi^Y(y)dy$$

$$+ \int_{\mathbb{R}^+} [-(\beta - \delta)(z_1 - z)^+ + (-\delta)(z - z_2)^+]\pi^Z(z)dz$$

where

$$(1)(\beta - \delta)z_1 = \lambda; \delta z_2 = \alpha - \lambda, \text{for } 0 \leq z_1 \leq \tfrac{\alpha}{\beta} \leq z_2 \text{ and } \lambda \in (0,\alpha), \delta \in (0,\beta);$$

$$(2)\lambda(K_3 - K_2) + \delta(K_3^b - K_2^b) = \alpha K_3 + \beta K_3^b - \gamma;$$

$$(3)K_1^b = z_1 K_2, K_2^b = z_2 K_1, K_3^b = z_1 K_4, K_4^b = z_2 K_3;$$

$$(4)0 \leq \lambda < \alpha; 0 \leq \delta < \beta;$$

$$(5)0 \leq K_1 \leq K_2 \leq K_3 \leq K_4.$$

For the problem in (A30), there exists a feasible solution by setting

$$z_1 \to 0; z_2 \to \infty; K_1 \to 0; K_4 \to \infty; \lambda = 0; \delta = 0;$$
$$K_2 = K_3 = K_a; z_2 K_1 = z_1 K_4 = K_b; z_1 K_2 \to 0; z_2 K_3 \to \infty; \tag{A31}$$

where $\alpha K_a + \beta K_b = \gamma$. This solution leads to upper bounds in Proposition 3. The objective function in (A30) is bounded above. Therefore, this program must have a solution which yields the least upper price bound on the basket option in (4).

(2) We now build up a class of pricing functions $p(x, y)$ which support these dominating portfolios. Define two separated sets as follows:

$$\begin{aligned} A = & (0, K_1] \times [K_2^b, K_3^b] \cup [K_1, K_2] \times [K_1^b, K_2^b] \cup [K_2, K_3] \times (0, K_1^b] \\ & \cup [K_2, K_3] \times [K_4^b, \infty) \cup [K_3, K_4] \times [K_3^b, K_4^b] \cup [K_4, \infty) \times [K_2^b, K_3^b], \\ B = & \mathbb{R}_+^2 \setminus A. \end{aligned} \tag{A32}$$

So the set $A$ indicates the region where dominating strategies exactly replicate a basket option, and the set $B$ is its complement.

When only dollar-denominated options are traded, a price function $p$ is attained, associated with two strikes $K_a^*$ and $K_b^*$, as shown in the left panel of Figure 1. Proposition 3 has established that

$$p(x, y) = \begin{cases} \geq 0, & \text{if } (x, y) \in (0, K_a^*] \times (0, K_b^*]; \\ \geq 0, & \text{if } (x, y) \in (K_a^*, \infty) \times (K_b^*, \infty); \\ 0, & \text{otherwise.} \end{cases}$$

If the $X$-denominated options on $Y$ are traded, their quoted prices in markets may be consistent with the price function $p$ so that valuation bounds are then not tightened. This scenario is linked to a feasible solution in (A30). Otherwise, we consider the following construction.

(I) Given the price function $p$, we first pick up two positive random numbers $z_1$ and $z_2$ so that $0 < z_1 < \tfrac{\alpha}{\beta} < z_2 < \infty$. Four points from $X$ are chosen so that

$$0 < K_1 < K_2 < K_a^* < K_3 < K_4,$$

and four points from $Y$ are then determined

$$K_1^b = z_1 K_2 < K_2^b = z_2 K_1 < K_b^* < K_3^b = z_1 K_4 < K_4^b = z_2 K_3.$$

Therefore, the joint density $p$ is divided into 36 small partitions.

To construct a density shown in the right panel of Figure 1, the density of one node $(x, y)$ in the set $A$ is added by a small number $\epsilon > 0$, and then the density of another node $(x, \tilde{y})$ or $(\tilde{x}, y)$ in the set $B$ is subtracted by $\epsilon$. In this way, a new price function $\hat{p}$ that is consistent with the marginals $\pi^X$ and $\pi^Y$ is constructed from the density $p$.

(II) Since the dollar-denominated options on $X$ and $Y$ are properly priced under the new price function $\hat{p}$, the marginal condition implies that:

$$\mathbb{E}_{\hat{p}}[X] = \int_{(x,y) \in A} x \hat{p}(x,y) dx dy = \int_{(x,xz) \in A} x \hat{p}(x,xz) dx dz = x_0.$$

For $(x, xz) \in A$ and $v_z \sim U[0,1]$, define $\int_x x \hat{p}(x, xz) dx = x_0 v_z$ so that $\int_z \int_x x \hat{p}(x, xz) dx dz = \int_z x_0 v_z dz = x_0$.

To be consistent with the marginal $\pi^Z$, a new price function $p^*$ is constructed from $\hat{p}$ so that $\int_x x p^*(x, xz) dx = x_0 \pi^Z(z)$ for $(x, xz) \in A$. This requires that $\int_x x(\hat{p}(x, xz) - p^*(x, xz)) dx = x_0(v_z - \pi^Z(z))$ and also $\int_z x_0(v_z - \pi^Z(z)) dz = 0$. We pick up three random numbers $z_a, z_b$ and $z_c$ from $Z$ so that $z_a < z_b = \sqrt{z_a z_c} < z_c$ and

$$v_{z_a} - \pi^Z(z_a) + v_{z_b} - \pi^Z(z_b) + v_{z_c} - \pi^Z(z_c) = 0.$$

For two small positive numbers $\kappa_1$ and $\kappa_2$ ($\kappa_1(v_{z_b} - \pi^Z(z_b)) > \kappa_2(v_{z_c} - \pi^Z(z_c))$), three points from $X$ in the set $A$ are chosen so that the adjustment of the density $\hat{p}$ at each node is reported as follows:

|       | $z_a$       | $z_b$                | $z_c$        |
|-------|-------------|----------------------|--------------|
| $x_a$ | $0$         | $-\kappa_1$          | $+\kappa_1$  |
| $x_b$ | $+\kappa_1$ | $\kappa_2 - \kappa_1$ | $-\kappa_2$  |
| $x_c$ | $-\kappa_2$ | $+\kappa_2$          | $0$          |

To be consistent with the density $\pi^Z$, all three points $x_a$, $x_b$ and $x_c$ ($x_b^2 = x_a x_c$) are attained by solving an equation system:

$$x_a = \frac{(v_{z_b} - \pi^Z(z_b))^2}{\kappa_1(v_{z_b} - \pi^Z(z_b)) - \kappa_2(v_{z_c} - \pi^Z(z_c))} x_0^2,$$

$$x_b = \frac{(v_{z_b} - \pi^Z(z_b))(v_{z_c} - \pi^Z(z_c))}{\kappa_1(v_{z_b} - \pi^Z(z_b)) - \kappa_2(v_{z_c} - \pi^Z(z_c))} x_0^2,$$

$$x_c = \frac{(v_{z_c} - \pi^Z(z_c))^2}{\kappa_1(v_{z_b} - \pi^Z(z_b)) - \kappa_2(v_{z_c} - \pi^Z(z_c))} x_0^2.$$

By repeating this process, we have $\int_z x_0(v_z - \pi^Z(z)) dz = 0$. A new price function $p^*$ is constructed from $\hat{p}$, associated with two adjustable variables $\kappa_1$ and $\kappa_2$. Hence, the $X$-denominated options on $Y$ are priced correctly under this price function.

Suppose there exists a group of parameters $(z_1^*, z_2^*, K_1^*, K_2^*, K_3^*, K_4^*)$ that support such a price function $p^*$. Given these parameters, two separated sets $A^*$ and $B^*$ are determined so that $A^* \cup B^* = \mathbb{R}_+^2$. Then there must exist a candidate joint distribution:

$$p^*(x,y) = \begin{cases} \geq 0, & \text{if } (x,y) \in A^*; \\ = 0, & \text{if } (x,y) \in B^*, \end{cases} \tag{A33}$$

so that $\int_{\mathbb{R}^+} p^*(x,y) dy = \pi^X(x)$, $\int_{\mathbb{R}^+} p^*(x,y) dx = \pi^Y(y)$ and $\int_{\mathbb{R}^+} x p^*(x, xz) dx = \pi^Z(z)/x_0$.

The quantities of $\lambda^*$ and $\delta^*$ are derived from the first condition in (A30). Therefore, the dominating strategy provides the least upper bound on this basket option

$$\mathbb{E}_{p^*}[g^*(x) + h^*(y) + xf^*(z)1_{z=y/x} - (\alpha x + \beta y - \gamma)^+] = 0.$$

This condition ensures the existence of a candidate joint density specified in (A33). $\square$

**Appendix F. Proof of Lemma 3**

**Proof.** First of all, define two new random variables by setting $\tilde{X} = \alpha X$ and $\tilde{Y} = \beta Y$ so that the payoff of the option **b** is re-expressed as $[\tilde{X} + \tilde{Y} - \gamma]^+$ for $(\alpha, \beta, \gamma) \in \mathbb{R}_3^+$. According to Hobson, Laurence and Wang (2005b) [23], partition $\mathbb{R}^+$ into $(2n + 1)$ $(n \geq 1)$ finite intervals in this way:

$$0 = \tilde{K}_0^1 < \tilde{K}_1^1 < \cdots < \tilde{K}_{2n}^1 < \gamma < \tilde{K}_{2n+1}^1 = \infty,$$

so that $(0, \tilde{K}_1^1) \cup (\cup_{i=1}^{2n} [\tilde{K}_i^1, \tilde{K}_{i+1}^1)) \cup [\tilde{K}_{2n+1}^1, \infty) = (0, \infty)$. Let

$$\tilde{K}_i^2 = \gamma - \tilde{K}_{2n+1-i}^1.$$

Hence, $\mathbb{R}_2^+$ may be expressed as the union of finite intervals:

$$\mathbb{R}_2^+ = \bigcup_{i,j=1}^{2n+1} R_{ij}, \text{ for } R_{i,j} = \{(\tilde{x}, \tilde{y}) \in \mathbb{R}_2^+ : \tilde{K}_{i-1}^1 \leq \tilde{x} < \tilde{K}_i^1, \tilde{K}_{j-1}^2 \leq \tilde{y} < \tilde{K}_j^2\}.$$

Now consider a portfolio which consists of two components:

(1) the holdings of the dollar-denominated options on $X$ are expressed as follows:

$$f_X(\tilde{x}) = \tilde{x}^+ + \sum_{i=1}^n \{(\tilde{x} - \tilde{K}_{2i}^1)^+ - (\tilde{x} - \tilde{K}_{2i-1}^1)^+\}; \tag{A34}$$

(2) the holdings of the dollar-denominated options on $Y$ are expressed as follows:

$$f_Y(\tilde{y}) = \tilde{y}^+ + \sum_{j=1}^n \{(\tilde{y} - \tilde{K}_{2j}^2)^+ - (\tilde{y} - \tilde{K}_{2j-1}^2)^+\}, \tag{A35}$$

associated with the amount of cash, $\omega = \sum_{l=1}^n (\tilde{K}_{2l}^2 - \tilde{K}_{2l-1}^2) - \gamma$.

At maturity, the payoff to this portfolio in each region $R_{ij}$ would be equal to

$$\begin{aligned}
f_X(\tilde{x}) + f_Y(\tilde{y}) + \omega &= f_X(\tilde{x}) - f_X(\gamma - \tilde{y}) \\
&= \tilde{x}^+ + \sum_{m=1}^{i \leq n} ((\tilde{x} - \tilde{K}_{2m}^1)^+ - (\tilde{x} - \tilde{K}_{2m-1}^1)^+) \\
&\quad - \{(\gamma - \tilde{y})^+ + \sum_{k=1}^{j \leq n} ((\tilde{K}_{2k-1}^2 - \tilde{y})^+ - (\tilde{K}_{2k}^2 - \tilde{y})^+)\},
\end{aligned} \tag{A36}$$

due to $f_X(\gamma - \tilde{y}) + f_Y(\tilde{y}) + \omega = 0$.

Since the function $f$ has the slope 0 or 1, the mean value theorem implies that

$$f_X(\tilde{x}) - f_X(\gamma - \tilde{y}) \leq [\tilde{x} - (\gamma - \tilde{y})]^+ = [\alpha x - (\gamma - \beta y)]^+ = [\alpha x + \beta y - \gamma]^+. \tag{A37}$$

In other words, this result may be re-expressed as

$$[f_X(\tilde{x}) - f_X(\gamma - \tilde{y})] - [\tilde{x} + \tilde{y} - \gamma]^+ = \\
\begin{cases}
0, & \text{if } \tilde{x} + \tilde{y} = \gamma; \\
0, & \text{if } \tilde{x} + \tilde{y} > \gamma \text{ and } \tilde{x}, \gamma - \tilde{y} \in (\tilde{K}_{2i}^1, \tilde{K}_{2i+1}^1); \\
0, & \text{if } \tilde{x} + \tilde{y} < \gamma \text{ and } \tilde{x}, \gamma - \tilde{y} \in (\tilde{K}_{2i-1}^1, \tilde{K}_{2i}^1); \\
< 0, & \text{otherwise.}
\end{cases} \tag{A38}$$

Therefore, the terminal payoff in (A36) sub-replicates the option **b** in the region $R_{ij}$, and thus it is dominated by the option **b** over $\mathbb{R}_2^+$. The initial cost provides a lower price bound on this option:

$$
\begin{aligned}
\underline{\mathcal{B}}(n) = &\int_{\mathbb{R}_+} \alpha \Big( x + \sum_{i=1}^n \big( (x - \tilde{K}_{2i}^1/\alpha)^+ - (x - \tilde{K}_{2i-1}^1/\alpha)^+ \big) \Big) \pi^X(x) dx \\
&- \int_{\mathbb{R}_+} \beta \Big( (\frac{\gamma}{\beta} - y)^+ + \sum_{j=1}^n \big( (\tilde{K}_{2j-1}^2/\beta - y)^+ - (\tilde{K}_{2j}^2/\beta - y)^+ \big) \Big) \pi^Y(y) dy.
\end{aligned}
\tag{A39}
$$

Let $K_i^1 = \tilde{K}_i^1/\alpha$ and $K_j^2 = \tilde{K}_j^2/\beta$ $(1 \le i, j \le 2n+1)$. To see that the bound $\underline{\mathcal{B}}(n)$ is the non-increasing function of partition number $n$. Recall the terminal payoff to the hedging portfolio:

$$
\begin{aligned}
&f_X(\alpha x) + f_Y(\beta y) + \omega = (-\alpha)(K_1^1 - x)^+ + \alpha(x - K_2^1)^+ + (-\beta)(K_{2n-1}^2 - y)^+ \\
&+ \beta(y - K_{2n}^2)^+ + (\alpha K_1^1 + \beta K_{2n}^2 - \gamma) + \alpha \sum_{i=2}^n \big( (x - K_{2i}^1)^+ - (x - K_{2i-1}^1)^+ \big) \\
&+ \beta \sum_{j=1}^{n-1} \big( (y - K_{2j}^2)^+ - (y - K_{2j-1}^2)^+ \big) + \beta \Big( \sum_{l=1}^{n-1} (K_{2l}^2 - K_{2l-1}^2) \Big) \\
&= (-\alpha)(K_1^1 - x)^+ + \alpha(x - K_2^1)^+ + (-\beta)(K_{2n-1}^2 - y)^+ + \beta(y - K_{2n}^2)^+ \le [\alpha x + \beta y - \gamma]^+,
\end{aligned}
\tag{A40}
$$

where $\alpha K_1^1 + \beta K_{2n}^2 = \alpha K_2^1 + \beta K_{2n-1}^2 = \gamma$, because for each $x \in [K_{2i-1}^1, K_{2i}^1)$, there exists a $y = \frac{\gamma - \alpha x}{\beta} \in [K_{2n+1-2i}^2, K_{2n+2-2i}^2)$ so that

$$
\begin{aligned}
&\alpha \sum_{i=2}^n \big( (x - K_{2i}^1)^+ - (x - K_{2i-1}^1)^+ \big) + \beta \sum_{j=1}^{n-1} \big( (y - K_{2j}^2)^+ - (y - K_{2j-1}^2)^+ \big) + \beta \Big( \sum_{l=1}^{n-1} (K_{2l}^2 - K_{2l-1}^2) \Big) \\
&= 0
\end{aligned}
\tag{A41}
$$

Now consider a strategy that involves selling $\alpha$ puts and buying $\alpha$ calls on $X$ with strikes $K_1^1$ and $K_2^1$, and selling $\beta$ puts and buying $\beta$ calls on $Y$ with strikes $K_{2n-1}^2$ and $K_{2n}^1$. This strategy sub-replicates the payoff $[\alpha x + \beta y - \gamma]^+$. In sum, the bound $\underline{\mathcal{B}}(n)$ is the decreasing function of partition number $n$.  □

## Appendix G. Proof of Proposition 6

**Proof.** Given a triplet $(\alpha, \beta, \gamma) \in \mathbb{R}_3^+$, Lemma 3 implies that sub-replicating strategies for the option **b** consist of two components:

(i) the holdings of the dollar-denominated options on $X$ involve short $\alpha$ puts at strike $K_a^1$ and long $\alpha$ calls at strike $K_a^2$;

(ii) the holdings of the dollar-denominated options on $Y$ involve short $\beta$ puts at strike $K_b^1$ and long $\beta$ calls at strike $K_b^2$,

where $\alpha K_a^1 + \beta K_b^2 = \alpha K_a^2 + \beta K_b^1 = \gamma$. It is easy to verify that the terminal payoffs generated by these strategies would sub-replicate the option **b** in this way:

$$
g(x) + h(y) - [\alpha x + \beta y - \gamma]^+ = \begin{cases} 0, & \text{if } (x,y) \in (0, K_a^1] \times [K_b^2, \infty) \text{ and } \alpha x + \beta y \ge \gamma; \\ 0, & \text{if } (x,y) \in [K_a^1, K_a^2] \times [K_b^1, K_b^2] \text{ and } \alpha x + \beta y \le \gamma; \\ 0, & \text{if } (x,y) \in [K_a^2, \infty) \times (0, K_b^1] \text{ and } \alpha x + \beta y \ge \gamma; \\ < 0, & \text{otherwise.} \end{cases}
\tag{A42}
$$

Given a triplet $(\alpha, \beta, \gamma) \in \mathbb{R}_3^-$, we consider trading strategies as follows

(i) the holdings of the dollar-denominated options on $X$ involve short $\alpha$ puts at strike $K_a^1$, long $\alpha$ calls at strike $K_a^2$, and long $\alpha$ calls and short $\alpha$ puts at strike $K_X$;

(ii) the holdings of the dollar-denominated options on $Y$ involve short $\beta$ puts at strike $K_b^1$, long $\beta$ calls at strike $K_b^2$, and long $\beta$ calls and short $\beta$ puts at strike $K_Y$,

where $\alpha K_a^1 + \beta K_b^2 = \alpha K_a^2 + \beta K_b^1 = \alpha K_X + \beta K_Y = \gamma$. It is easy to verify that the terminal payoffs generated by these strategies would sub-replicate the option **b** as shown in (A42). Therefore, the valuation bounds on the option **b** can be represented in (17) and (18). The solution sets of these programs are not empty, provided that there always exist strikes for $K_a^1$, $K_a^2$, $K_b^1$, $K_b^2$, $K_X$ and $K_Y$. Also, these programs are bounded from above and thus must have solutions.

To see that there exists a price function $p$ that supports the bound in (17) or (18), consider the construction of a counter-monotonic process $(X, Y)$ so that the variable $Y$ is a non-increasing function of the variable $X$. More specifically, there exists a real number vector $(\bar{x}, \bar{y}) \in \mathbb{R}_2^+$ so that

$$prob(X \le \bar{x}) = v; prob(Y \le \bar{y}) = 1 - v, \text{ for } v \sim U[0,1]. \tag{A43}$$

Then a counter-monotonic process $(X, Y)$ is constructed through the inverse function, and the variables $X$ and $Y$ have the desired marginal densities. The joint density of $(X, Y)$ is specified in (19). □

**Appendix H. Proof of Proposition 7**

**Proof.** (1) Given a triplet $(\alpha, \beta, \gamma) \in \mathbb{R}_3^+$, we shall set up a class of dominated hedging portfolios for the option **b**. First of all, pick up four points from $X$ and four points from $Y$:

$$0 < K_1 < K_2 < K_3 < K_4; 0 < K_1^Y < K_2^Y < K_3^Y < K_4^Y, \tag{A44}$$

so that there exists two variables $z_1$ and $z_2$ $(0 < z_1 \le z_2 < \infty)$

$$\text{i) } z_1 = \frac{K_1^Y}{K_3} = \frac{K_2^Y}{K_4}, z_2 = \frac{K_3^Y}{K_1} = \frac{K_4^Y}{K_2}, \tag{A45}$$

and

$$\text{ii) } \alpha K_1 + \beta K_3^Y = \gamma; \alpha K_2 + \beta K_2^Y = \gamma; \alpha K_3 + \beta K_1^Y = \gamma. \tag{A46}$$

To sub-replicate the payoff **b**, consider a trading strategy:

(i) the holdings of the dollar-denominated options on $X$ consist of long $\lambda$ calls at strike $K_3$ and short $(\lambda - \alpha)$ call at strike $K_4$;
(ii) the holdings of the dollar-denominated options on $Y$ consist of long $\delta$ calls at strike $K_3^Y$ and short $(\delta - \beta)$ call at strike $K_3^Y$;
(iii) the holdings of the $X$-denominated options on $Y$ consist of short $\beta$ puts at strike $z_1$, and short $(\delta - \beta)$ calls at strike $z_2$,

for $\lambda \ge \alpha$ and $\delta \ge \beta$.

Following the conditions in (20),

$$\beta z_1 = \lambda - \alpha; (\delta - \beta)z_2 = \alpha, \text{ for } 0 < z_1 \le z_2 < \infty;$$
$$\lambda(K_4 - K_3) + \delta(K_4^Y - K_3^Y) = \alpha K_4 + \beta K_4^Y - \gamma, \tag{A47}$$

we can identify the following equalities, associated with (A45) and (A46):

$$\lambda K_3 = \delta K_3^Y = \gamma;$$
$$(\lambda - \alpha)K_4 + (\delta - \beta)K_4^Y = \gamma. \tag{A48}$$

With these equalities, the strategy identified above sub-replicates the option **b** in the following way

$$g(x) + h(y) + xf(z)1_{z=y/x} - [\alpha x + \beta y - \gamma]^+ =$$

$$\begin{cases} 0, & \text{if } 0 < x \le K_3 \text{ and } 0 < y \le K_3^Y \text{ for } \frac{y}{x} \in [z_1, z_2] \text{ and } \alpha x + \beta y \le \gamma; \\ 0, & \text{if } 0 < x \le K_2 \text{ and } K_3^Y < y \le K_4^Y \text{ for } \frac{y}{x} \ge z_2 \text{ and } \alpha x + \beta y \ge \gamma; \\ 0, & \text{if } K_3 < x \le K_4 \text{ and } 0 < y \le K_2^Y \text{ for } \frac{y}{x} \le z_1 \text{ and } \alpha x + \beta y \ge \gamma; \\ 0, & \text{if } x > K_4 \text{ and } y > K_4^Y \text{ for } \frac{y}{x} \in [z_1, z_2]; \\ < 0, & \text{otherwise.} \end{cases} \quad (A49)$$

Note that since there always exit the strikes $K_i$ and $K_i^Y$ ($i = 1, 2, 3$) from the construction above, these strikes then determine the strikes $z_1$ and $z_2$ from the condition (A45). As a result, the strikes $K_4$ and $K_4^Y$ are also obtained from that condition. Hence, all the relevant strikes are determined once the strikes $K_i$ and $K_i^Y$ ($i = 1, 2, 3$) are chosen. This further implies that a dominated trading strategy identified above always exists. Moreover, the strategies identified in (17) can be viewed as the particular cases with the zero holdings of the $X$-denominated options on $Y$ for $z_1 \to 0$ and $z_2 \to \infty$.

Therefore, the lower valuation bounds on the option **b** are attained as follows:

$$\max_{\lambda, \delta, z_1, z_2, K_i, K_i^y, i=1,2,3,4} \int_{\mathbb{R}^+} (\lambda (x - K_3)^+ + (\alpha - \lambda)(x - K_4)^+) \pi^X(x) dx$$

$$+ \int_{\mathbb{R}^+} (\delta (y - K_3^Y)^+ + (\beta - \delta)(y - K_4^Y)^+) \pi^Y(y) dy \quad (A50)$$

$$+ \int_{\mathbb{R}^+} ((-\beta)(z_1 - z)^+ + (\beta - \delta)(z - z_2)^+) \pi^Z(z) dz,$$

s.t.

$$(1) \beta z_1 = \lambda - \alpha; (\delta - \beta) z_2 = \alpha, \text{ for } 0 \le z_1 \le z_2;$$
$$(2) \lambda (K_4 - K_3) + \delta (K_4^Y - K_3^Y) = \alpha K_4 + \beta K_4^Y - \gamma;$$
$$(3) \alpha K_1 + \beta K_3^Y = \gamma; \alpha K_3 + \beta K_1^Y = \gamma;$$
$$(4) \lambda \ge \alpha; \delta \ge \beta; 0 \le K_1 \le K_2 \le K_3 \le K_4.$$

According to the analysis above, this problem always has a feasible solution and it is bounded above. Thus it must have a solution. Since the strategies identified in (17) are the special cases of $f(z) \equiv 0$, it is equivalent to the case of $K_a^1 \to 0$ and $K_b^1 \to 0$ and thus the greatest lower bound derived from this problem should not be cheaper than the bound attained in (17).

The rest of the proof is about the construction of the pricing function that supports the greatest lower bound. When only dollar-denominated options are traded, a price function $p$ is attained and the strikes $\hat{K}_a^1$, $\hat{K}_a^2$, $\hat{K}_b^1$ and $\hat{K}_b^2$ are attained, as shown in the left panel of Figure 2. Under this pricing function, the dollar-denominated options on $X$ and $Y$ are priced correctly. However, this function is inconsistent with the prices of the $X$-denominated options on $Y$.

Given the pricing function $p$, we firstly pick up two positive numbers $z_1 = \frac{\hat{K}_b^1}{\hat{K}_a^2}$ and $z_2 = \frac{\hat{K}_b^2}{\hat{K}_a^1}$. Then one can draw the lines through these numbers to divide the space into three regions. Furthermore, another four points are chosen in this way:

$$\alpha K_x^2 + \beta K_y^2 = \gamma; K_x^4 = \frac{K_y^2}{z_1}; K_y^4 = z_2 K_x^2,$$

where $K_x^2 \in (\hat{K}_a^1, \hat{K}_a^2)$. Following the first procedure in the proof of Proposition 4, one can construct a pricing function $\hat{p}$ that is consistent with dollar-denominated options. To be consistent with cross-currency options, one follow the second procedure to construct a density function $p^*$ by adjusting $K_x^2$ and $K_y^2$.

Suppose there exists a group of parameters $(z_1, z_2, \hat{K}_a^1, \hat{K}_x^2, \hat{K}_a^2, \hat{K}_x^4)$ that support the following pricing function $p^*$:

$$
p^*(x, y) = \begin{cases}
\geq 0, & \text{if } (x, y) \in (0, \hat{K}_a^2] \times (0, z_2 \hat{K}_a^1] \text{ such that } \frac{y}{x} \in [\hat{z}_1, \hat{z}_2] \text{ and } \alpha x + \beta y \leq \gamma; \\
\geq 0, & \text{if } (x, y) \in (0, \hat{K}_x^2] \times [z_2 \hat{K}_a^1, z_2 \hat{K}_x^2] \text{ such that } \alpha x + \beta y \geq \gamma; \\
\geq 0, & \text{if } (x, y) \in [\hat{K}_a^2, \hat{K}_x^4] \times (0, z_1 \hat{K}_x^4] \text{ such that } \alpha x + \beta y \geq \gamma; \\
\geq 0, & \text{if } (x, y) \in [\hat{K}_x^4, \infty) \times [z_2 \hat{K}_x^2, \infty) \text{ such that } \frac{y}{x} \in [\hat{z}_1, \hat{z}_2]; \\
0, & \text{otherwise.}
\end{cases}
$$

One can construct a dominated trading strategy, while the quantities of $\lambda$ and $\delta$ are determined in the first condition of the problem (A50). With this pricing function, there is the following relation:

$$
\mathbb{E}_{p^*}[g^*(x) + h^*(y) + x f^*(z) 1_{z=y/x} - [\alpha x + \beta y - \gamma]^+] = 0.
$$

(2) For a triplet $(\alpha, \beta, \gamma) \in \mathbb{R}_3^-$, the analysis can be completed in the similar way. First of all, we have four points from $X$ and four points from $Y$:

$$
0 < K_1 < K_2 < K_3 < K_4; 0 < K_1^Y < K_2^Y < K_3^Y < K_4^Y, \tag{A51}
$$

so that

$$
\begin{aligned}
& K_1^Y = z_1 K_3, K_2^Y = z_1 K_4, K_3^Y = z_2 K_1, K_4^Y = z_2 K_2, \text{for } 0 < z_1 \leq z_2 < \infty, \\
& \alpha K_1 + \beta K_3^Y = \gamma; \alpha K_2 + \beta K_2^Y = \gamma; \alpha K_3 + \beta K_1^Y = \gamma.
\end{aligned} \tag{A52}
$$

Following the conditions in (20)

$$
\begin{aligned}
& \beta z_1 = -\lambda; \delta z_2 = -\alpha, \text{for } 0 < z_1 \leq z_2 < \infty; \\
& \lambda(K_3 - K_4) + \delta(K_3^Y - K_4^y) = \alpha K_3 + \beta K_3^Y - \gamma,
\end{aligned} \tag{A53}
$$

we can identify the following equalities, associated with (A52) and (A46):

$$
\begin{aligned}
& (-\alpha)(K_3 - K_2) + \lambda(K_3 - K_4) = 0; \\
& (-\beta)(K_3^Y - K_2^Y) + \delta(K_3^Y - K_4^Y) = 0.
\end{aligned} \tag{A54}
$$

With these equalities, the strategy identified in (21) sub-replicates the option **b**. Following the same argument (construction), there exists a joint density that supports the greatest lower valuation bound on this option. □

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
