# Peer review of "Arbitrage Bounds on Currency Basket Options"

_mca, doi:10.3390/mca25030060_

Round 1

Reviewer 1 Report

The paper considers the problem of deriving upper bounds for basket options on currencies. The paper is well-written and the results discussed in this paper are worth publishing. 

My main concern is that the authors claim their paper extents the paper of Hobson et al. However, that paper considers a basket size of n, whereas in the current version only X, Y and Z are present. 

Author Response

Thanks for the reviewer's valuable and constructive comments.

Author Response

(The authors gave the same response as above.)

Round 2

Reviewer 2 Report

I am happy with the author's response. The current version of the paper can be published in MCA.